

# An ice-sheet modelling framework for leveraging sub-ice drilling to assess sea level potential applied to Greenland

Benjamin A. Keisling1, Joerg M. Schaefer2,3, Robert M. DeConto4, Jason P. Briner5, Nicolás E. Young2, Caleb K. Walcott5, Gisela Winckler2,3, Allie Balter-Kennedy2,3, Sridhar Anandakrishnan6

[1]University of Texas Institute of Geophysics, Austin, TX

[2]Lamont-Doherty Earth Observatory, Palisades, NY

[3]Department of Earth and Environmental Sciences, Columbia University, New York, NY

[4]Department of Geosciences, University of Massachusetts Amherst, Amherst, MA

[5]Department of Geology, University at Buffalo, Buffalo, NY

[6]Department of Geosciences, Pennsylvania State University, University Park, PA

*Correspondence to*: Benjamin Keisling (keisling@ig.utexas.edu)

**Abstract.** The contribution of the Greenland Ice Sheet (GIS) to sea level rise (SLR) is accelerating and there is an urgent need to improve predictions of when and from what parts of the ice sheet Greenland will contribute its first meter. Estimating the volume of Greenland ice that was lost during past warm periods offers a way to constrain the ice sheet's response to future warming. Sub-ice sediment and bedrock, retrieved from deep ice core campaigns or targeted drilling efforts, yield critical and direct information about past ice-free conditions. However, it is challenging to scale the few available sub-ice point measurements to the geometry of the entire ice sheet. Here, we provide a framework for assessing sea-level potential, which we define as the amount the GIS has contributed to sea level when a particular location in Greenland is ice-free, from an ensemble of ice-sheet model simulations representing a wide range of plausible deglaciation scenarios. An assessment of dominant sources of uncertainty in our paleo ice sheet modelling, including climate forcing, ice-sheet initialization, and solid-Earth properties, reveals spatial patterns in the sensitivity of the ice sheet to these processes and related feedbacks. We find that the sea-level potential of central Greenland is most sensitive to lithospheric feedbacks and ice-sheet initialization, whereas the ice-sheet margins are most sensitive to climate forcing parameters. Our framework allows us to quantify the local and regional uncertainty in sea-level potential, which we use to evaluate the GIS bedrock according to the usefulness of information sub-ice sediments and bedrock provide about past ice-sheet geometry. Through our ensemble approach, we can assign a plausible range of GIS contributions to global sea level for deglaciated conditions at any site. Our results identify primarily areas in southwest Greenland, and secondarily north Greenland, as best-suited for subglacial access drilling that seeks to constrain the response of the ice sheet to past and future warming.



## 1 Introduction

Sea-level rise (SLR) is one of the most profound economic, social and environmental issues facing humanity. Flooding associated with ongoing SLR is projected to cost up to 3% of global gross domestic product annually by the end of this century if emissions continue unabated (Jevrejeva et al. 2018). In the United States, SLR disproportionately impacts communities of

color and those in low-income areas, exacerbating issues of environmental justice (Hardy et al. 2017). Globally, the displacement of hundreds of millions of people will have cascading social, political, and environmental impacts as populations in low-lying areas, especially in the developing world, are forced inland by rising seas (Geisler & Currens 2017). Accurately predicting the source of future SLR is critical to adaptation, because the spatial pattern of ice loss impacts that of SLR (Larour et al. 2017, Hamlington et al. 2020).


The rate of global SLR has nearly tripled since 1890 and continued to accelerate since satellite-based methods became widely used in the 1970s (Hay et al. 2015, Nerem et al. 2018). The relatively modest rates of SLR in the 19[th] and most of the 20[th] centuries were driven primarily by increased oceanic heat uptake of anthropogenic warming (Hay et al. 2015) and retreating mountain glaciers (Oerlemans 1994). In the last half-century, SLR accelerated as glacier retreat increased globally (Hugonnett

et al. 2021). In the last two decades, Greenland ice loss has emerged as a dominant driver of SLR (Mouginot et al. 2019, Coulson et al. 2022), very likely as the result of human perturbations to the global climate system (IPCC SROCC, 2019). Greenland is predicted to remain the single greatest contributor to SLR over the next half-century (Hanna et al. 2024), motivating the questions of *when, at what rate, and from where* will the next meter of global SLR from Greenland come?

Responses of the GIS to past periods of naturally forced global warmth may augur its future behaviour (Briner et al. 2020). The large-scale response of the ice sheet to warm climate during the Pleistocene (the last ~2.65 million years) has been the subject of ongoing debate, with some lines of evidence suggesting prolonged resilience and others indicating repeated retreat. Interglacial ice preserved at the base of the NEEM ice core in Northwest Greenland (Figure 1) indicates that during the Eemian (Marine Isotope Stage (MIS) 5e, 125 thousand years ago (ka)) the ice sheet remained extensive and continuous across the

island despite >8°C of warming at that site (NEEM Community Members 2013). Reyes (2014) argued that during MIS 11, while a portion of the southern GIS diminished, the ice sheet lost no more than 30-40% of its volume. Argon isotopes in the basal ice of the southern GIS (Dye-3; Figure 1) reinforce this interpretation and demonstrate that some basal ice there is as old as $400 \pm 170$ ka (Yau et al. 2016). The same approach yielded a basal ice age of $970 \pm 140$ ka for GRIP ice beneath the summit of the ice sheet (Yau et al. 2016). These findings are in accord with an analysis of the basal material from the GISP2 ice core

(Figure 1), which found that site had been continuously covered by ice for the last 2.7 Ma (Bierman et al. 2014). These results, which argue for the stability of the GIS in more-or-less its present configuration for the duration of the Pleistocene, are unsurprising in the context of previous modelling results. DeConto et al. (2008) demonstrated the sensitivity of GIS glaciation to atmospheric $CO_2$ levels, which have been near or below 280 ppm deglaciation threshold for at least the past 3 million years



(Bereiter et al 2015, Martinez-Boti et al. 2015); more recent studies have also demonstrated that a perturbation of short enough
magnitude and/or duration (e.g. < 1ka) cause changes that are broadly reversible, and do not lead to Greenland's large-scale deglaciation (Bochow et al. 2023).

In contrast, an emerging line of direct observations from the analysis of subglacial materials recovered via ice-core and sub-ice drilling has revolutionized the study of past ice-sheet stability. In Antarctica, studies have focused on the reconstruction of
ice-sheet thickness from interior sites using a so-called "dipstick" approach (Halberstadt et al. 2023, Jones et al. 2015), by determining the surface exposure history of nunataks that stick out of the ice-sheet today or at some time in the past. Long-archived basal rock and sediment from GISP2, at the centre of the ice sheet, and Camp Century, closer to the margins, are yielding new insight into ice-sheet stability through the Pleistocene. Cosmogenic-nuclide dating (Schaefer et al. 2016), multi-proxy analysis (Christ et al. 2021), and optically stimulated luminescence dating (Christ et al. 2023) in these archives provide
direct evidence that the ice sheet was absent from the site within a particular window of time; as these methods continue to develop and more samples become available, these windows are narrowing. Balter-Kennedy et al. (2021) added yet another dimension to the possibilities afforded by subglacial bedrock samples, showing that down-core cosmogenic nuclide measurements (e.g. into the bedrock) can yield insights to the total amount of erosion at a particular location, and thus a time-integrated history of ice dynamics (e.g. basal thermal state and basal velocity). Such breakthroughs, combined with a range of
other data sources including bedrock elevation, ice dynamics, ice surface conditions, subglacial lithology, and subglacial thermal state, are currently being used to plan the acquisition of subglacial samples that promise new insights into the ice-sheet history (Briner et al. 2022). But what does this knowledge of a particular location being ice-free tell us about the larger GIS vulnerability, and in turn, global sea level?

Global to regional sea-level reconstructions have placed constraints on past Pleistocene sea-level highstands (e.g. Dutton et al. 2015, 2021), although disentangling the relative contributions of specific ice sheets is difficult when relying on these indicators (Hay et al. 2014, Vyverberg et al. 2018, Barnett et al. 2023), particularly for magnitudes of sea level rise on the order of a few meters (Khan et al. 2017, Dyer et al. 2021). Paleo ice-sheet modelling can provide a complementary method for predicting where the first meter of sea level will originate, but differences between modelling approaches and in particular model forcing
makes it difficult to assess with high confidence which parts of the GIS margin are the most responsive to warming and at which time (Plach et al. 2018). Previous studies have used ice-sheet models to quantify the response of the GIS to specific periods of past warmth, and come to different conclusions about the resilience (Helsen et al. 2013) and geometry (e.g. Helsen et al. 2013, Stone et al. 2013, Robinson et al. 2011) of the ice sheet, even for the same Pleistocene interglacial (e.g. MIS 5e, Plach et al. 2018).


Here, we outline a novel approach using an ensemble of ice-sheet model simulations, not tied to a particular interglacial but rather encompassing a range of past warming scenarios, to define sea-level potential throughout Greenland. We apply this



methodology across Greenland to reveal regional patterns in the sensitivity of the ice sheet to different processes and feedbacks, and demonstrate how this information can be used to guide sub-ice drilling efforts to gain the greatest insights into past sea
level. In documenting the utility of this methodology, we also suggest ways forward for incorporating yet additional sources of uncertainty and providing data-based constraints that will further strengthen the inference of sea-level potential in the future.

## 2 Methods

Previous studies have documented a range of potential ice sheet responses to the same periods of past warmth, resulting in differing assessments about the resilience and geometry of the ice (Helsen et al. 2013, Stone et al. 2013, Robinson et al. 2011).
Differences in the modelled footprint of the ice sheet result primarily from differences in the experimental design, and in particular, the approach taken to climate forcing (Plach et al. 2018). However, the relative role of different processes and forcings, including uncertainty in surface mass balance, solid-Earth feedbacks, and ice-sheet initialization has not been fully assessed (Edwards et al. 2014), and is especially lacking at regional scale. Our methodology provides a framework for answering questions motivated by the recovery of subglacial materials from specific regions (e.g. Christ et al. 2023). Given
that we know a location was ice-free at a certain time, this framework constrains how much mass the ice sheet must have lost for that to happen, providing both a robust lower-limit and a range of plausible ice volume loss numbers. The approach allows us to pose, and answer, critical questions about the future of the ice sheet with a novel perspective, such as "what are the most likely source regions for the first meter of sea-level rise from Greenland?" Next, we describe our modelling approach and the decisions made with respect to each of the ensemble parameters (Table 1).

| Parameter | Value |
|---|---|
| Spatial climatology pattern | Preindustrial, Holocene Thermal Max |
| Precipitation Lapse Rate | 0, 2% ℃$^{-1}$ |
| Rate of interglacial warming | 1, 1.33, 1.67, 2 ℃ kyr$^{-1}$ |
| Lithospheric relaxation time | 500, 3000 yr |
| Starting geometry | Modern cold start, modern transient, LGM |

Table 1. Summary of parameters used to define the ensemble.

We have framed this study in terms of five model parameters that have been established as having a major impact on deglaciation processes across a range of timescales, and thus represent the major uncertainties in paleo ice-sheet reconstruction.
These are: spatial climatology pattern, warming rate, initial ice-sheet geometry, precipitation lapse-rate, and lithospheric relaxation time. We aim to establish end-members for each parameter that illustrate the main sources of uncertainty in defining sea-level potential while leaving room for future efforts to explore the parameter space in finer detail (Table 1). Two major





factors drive temperature change over Greenland on multi-centennial (i.e. deglacial) timescales: $CO_2$ (Buizert et al. 2018) and insolation (Helsen et al. 2013), resulting in different patterns and magnitudes of warmth; yet our knowledge of the distribution of past temperature change is limited, particularly for past warm periods. To encompass a range of spatial temperature patterns, we use two different starting climatologies representative of warmth driven by greenhouse gases (pre-industrial) versus high boreal summer insolation (Holocene Thermal Maximum; HTM), which are well-constrained from a synthesis of ice-core, observational, and model-derived outputs (Buizert et al. 2018; Figure 1). Temperature change can be rapid or slow; we use four rates of temperature change to account for this. Deglaciation may begin from an ice-sheet configuration that looks similar to modern or it may have continued straight out of a glacial maximum with the ice margin advanced to the continental shelf (Funder et al 2011); we use three different starting geometries to account for this. We also examine the impact on deglaciation patterns of increasing precipitation as a function of warming and different lithospheric response times, which determine the rate of glacial isostatic adjustment. Through our ensemble approach, we examine the response of the ice sheet to many different scenarios, and integrate the responses to constrain the sea-level potential of any particular site (defined as the amount the GIS has contributed to sea level when a particular location in Greenland is ice-free). For the error propagation, we encapsulate multiple sources of uncertainty by capturing different end-members in the climate forcing, initialization, and solid-Earth model, thereby placing uncertainties on estimates of sea-level potential that stem from these unknowns. Each unique set of parameters is subject to four different rates of atmospheric warming, allowing us to capture how uncertainty in these parameters affects the way the ice sheet retreats under diverse warming scenarios (Table 1). We map the GIS response to warming, in order to (1) estimate of the region(s) of GIS that are most likely to contribute to the first meter of global sea-level change, (2) guide future sub-glacial access efforts that can provide targeted information about the response of the ice sheet to past warming, and (3) contextualize existing and future datasets within a glaciologically coherent, full-geometry framework to establish the minimum GIS contribution to past sea level when a particular location is ice-free (e.g. Christ et al. 2023). From our resulting map, we can infer the range of plausible sea-level potential of any part of the GIS, regardless of when the most recent deglaciation occurred. Stated differently, the map illustrates which segments of the GIS are most vulnerable under diverse climatic warming scenarios.





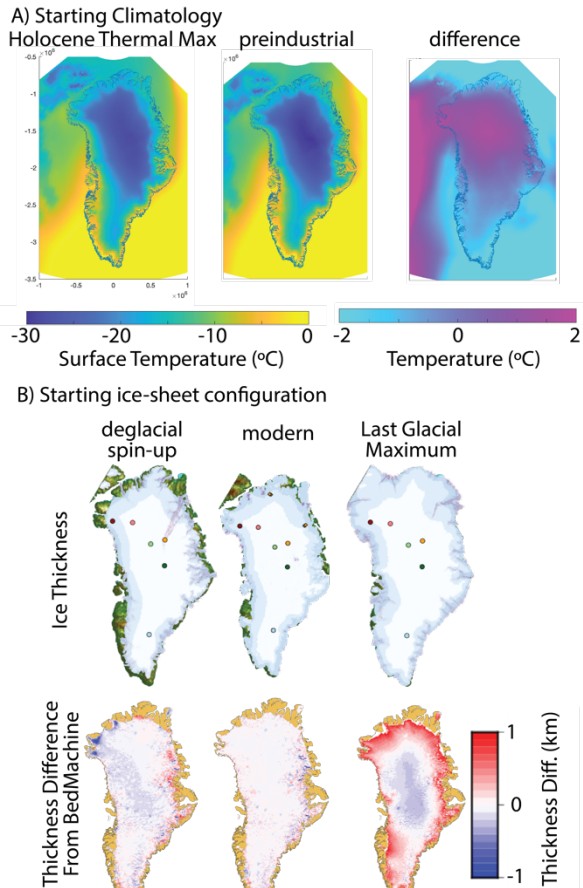

**Figure 1. Ice sheet model forcing and initialization. A) Two climatologies are used to initialize the climate forcing. The first is from**
**the Holocene Thermal Maximum, and the second is modern (preindustrial). The difference between the two climatologies shows**
**that the HTM climate is warmer in North and West Greenland by up to 2ºC. B) Three starting ice-sheet configurations are used in**
**the ensemble: a modern ice sheet spun up by running the model through a glacial cycle, the ice sheet is initialized based on modern**
**thickness observations (Morlighem et al. 2017), and a modelled Last Glacial Maximum ice sheet. Thickness differences are shown**
**relative to BedMachine (Morlighem et al. 2017). Coloured dots indicate deep ice-core sites discussed in the text: Camp Century**
**(Deep Red), NEEM (Pink), NGRIP (light green), EGRIP (orange), GRIP/GISP2 (dark green), Dye-3 (light blue).**

Figure 2 shows the method that we apply to calculate sea-level potential, and the sensitivity of the sea-level potential to each
of our ensemble parameters. For each 10 km model grid cell, we analysed the ensemble to find the first timestep the site
becomes ice-free in each simulation (Figure 2a). For the first ice-free timestep in each simulation, we save the ice-sheet volume
and extent, and collate these parameters across the whole ensemble (Figure 2b,c). Repeating this process for each grid cell, we
also look at our results along every dimension of our ensemble, enabling us to calculate the importance of each parameter for
each site and rank which of the considered parameters dominate the ensemble spread at each location. We define the sensitivity



to each parameter as the width of the ensemble spread for each parameter separately divided by the width of the full ensemble.
The parameter that most reduces the spread of 'Greenland contribution to SLE (m)' (x-axis of Figure 1b) indicates the
parameter a given region is most sensitive to. The lower the sensitivity, the more that knowledge about that parameter would
reduce uncertainty in sea-level potential for that site (Figure 2b).

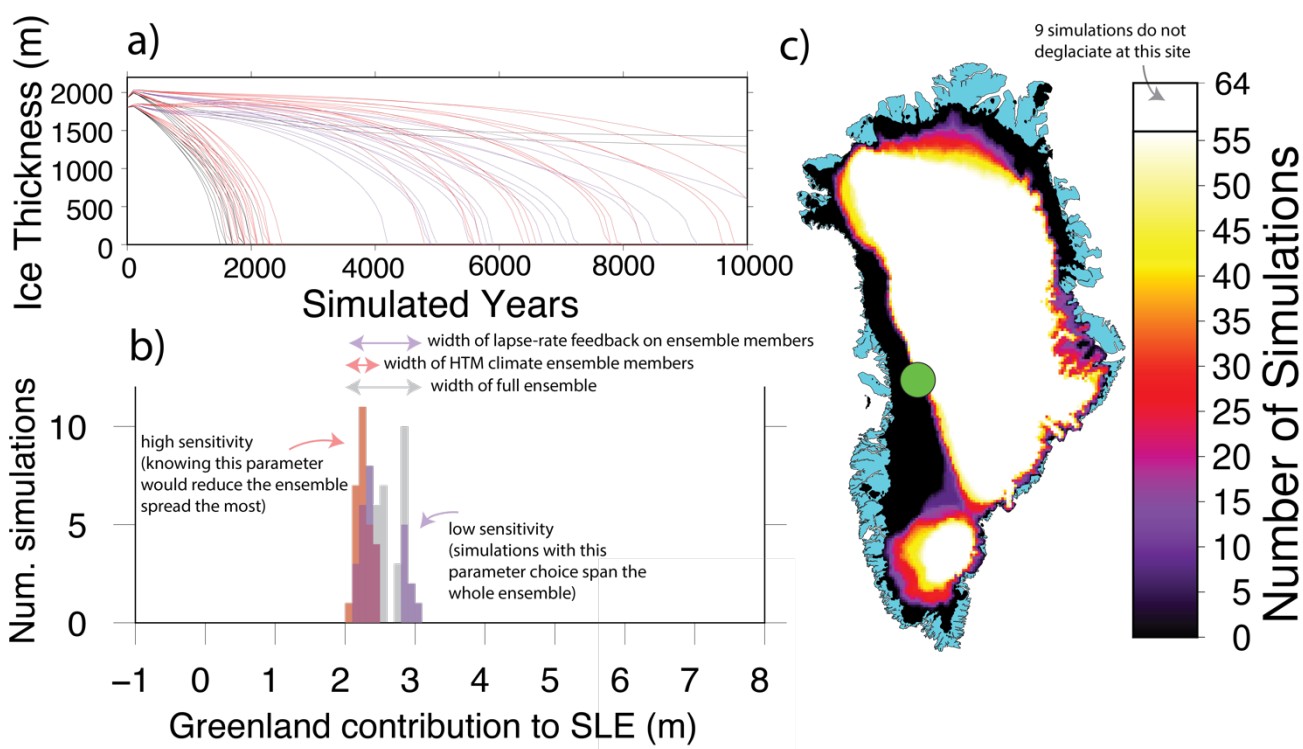


**Figure 2: Ensemble design. An example of our results shown for one location in West Greenland. A) Ice thickness at the green dot
in panel C) plotted for all ensemble members. Each simulation is represented by one thin line. Simulations that reach thickness=0
at some point during the deglaciation are used to calculate sea-level potential for this site. Purple and red lines correspond to purple**
**and red histograms in panel B. B) Histogram of outcomes for the location shown with the green dot in panel C. The contribution of
Greenland to global sea level when this site becomes ice-free ranges from 2.0 meters to 3.2 meters. The ensemble members which all
have the precipitation lapse rate turned off are superimposed on the histogram in purple. The ensemble members with a HTM
climatology are superimposed in red. This site is most sensitive to HTM climate, because knowing that parameter with certainty
would reduce the spread of the ensemble by the greatest amount. C) Greenland footprint associated with ice-free conditions for the**
**location in West Greenland identified with a green dot. Black regions indicate that every simulation is ice-free at the same time that
this location deglaciates, whereas white regions are still ice-covered in every simulation when this location becomes ice-free.**



### 2.1 Ice Sheet Model

We used a three-dimensional thermomechanical ice sheet model that uses a hybrid ice-flow law that efficiently bridges between

fast-flowing areas of streaming ice (Shallow Shelf Approximation) and inland areas of low velocity and high driving stress (Shallow Ice Approximation) (Pollard & DeConto 2012a). The model has been validated against other ice-sheet models under a range of conditions (e.g. Cornford et al. 2020, Pollard & DeConto 2020) and has been used extensively for paleo and future ice-sheet simulations in Antarctica (DeConto et al. 2021) and the Northern Hemisphere (Han et al. 2021). The model uses a Weertman-type sliding law for basal ice motion with basal sliding coefficients calculated through an inverse scheme that

iteratively adjusts sliding to reduce the mismatch between the modelled and observed ice-sheet geometry for the present-day ice sheet (Pollard & DeConto 2012b). The model has been applied to understand paleoclimate scenarios where both the boundary conditions and model forcing differ substantially from modern-day. It is therefore well suited for the relatively long integrations run here, and even longer (100ka–1Ma) integrations that offer complementary ways to study the evolution of the ice sheet on glacial-interglacial timescales in future work.

### 2.2 Ensemble Design

We ran ninety-six simulations varying four key parameters: starting climatology, lapse rate for precipitation, aesthenosphere relaxation time, and starting geometry. Each combination of parameters was subject to four different rates of glacial-to-interglacial warming applied for 10,000 years (see Section 2.2.3). In analysing the ensemble, we focus on one set of experiments initialized with a modern geometry and one set of experiments initialized with an LGM geometry so as not to

weight the results toward modern, resulting in 64 simulations evaluating sea-level potential for each 10km by 10km grid box. This approach and its implications are discussed in greater detail in Section 2.2.5.

### 2.2.1 Initial Climate Forcing

A primary control on the spatial pattern of GIS deglaciation is the pattern of surface mass balance (SMB). SMB reconstructions exist continuously for the last 21 kyr, because during this period ice cores, climate models and modern observational data

overlap (Buizert et al. 2018). Conversely, for most past warm periods before 21 ka there is little known about the precise patterns of SMB. Thus, we select two representative time periods from the Holocene to represent end-members in the SMB forcing (Figure 1). First, we select a time slice in the early Holocene at 8.5 ka when warm (especially summer) temperatures were driven by Earth's orbital configuration, resulting in a more developed ablation zone in northern Greenland, and a reduced ablation zone in western Greenland. The second time-slice chosen is pre-industrial (1850 CE), when warm annual temperatures

were driven by increased atmospheric $CO_2$, resulting in minimal melting in northern Greenland and a well-developed ablation zone in western Greenland. Both forcings come from a blended model-data reconstruction that includes seasonally resolved spatial and temporal variability (Buizert et al. 2018) downscaled to 40km resolution (Figure 1).



### 2.2.2 Lapse rate applied to precipitation

Ice-sheet margin retreat inland of its present-day position during past warm periods can be volumetrically by increased
accumulation in precipitation-limited areas whose temperature remains far below the melting point. As climate warms, the
atmosphere's capacity to hold water is enhanced, which can lead to increasing precipitation rates (and thus ice sheet thickening)
for inland ice-sheet regions (Payne et al. 2021). We account for this by considering a precipitation correction that increases
precipitation by 2% per degree of temperature increase in each grid cell. This "precipitation-lapse rate correction" enables us
to consider the impact of the feedback between a warming atmosphere and its moisture content/capacity in calculating ice-
volume changes. We consider this value to be a plausible upper-bound, as it has been found to accurately reproduce glacial-
interglacial changes in precipitation rate (Ritz et al. 2001, Abe-Ouchi et al. 2007). The lower-bound on SMB is determined by
not applying the precipitation lapse-rate correction, because there is no compensation for increasing melt.

### 2.2.3 Rate of interglacial warming

The GIS volume decreased in response to past variations in natural forcing, including orbital changes, changes in ocean
circulation, and atmospheric greenhouse gasses. Mechanisms driving past climate warming vary among interglacial periods
(PAGES 2016) and the timing of ice-sheet retreat during pre-LGM interglaciations is only coarsely constrained (Schaefer et
al. 2016), so we leverage the best-studied periods of past ice-sheet retreat to understand possible rates of interglacial climate
warming. During the last deglaciation, Greenland's mean annual temperature increased by ~18°C between 18ka and 12ka, an
average rate of 3°C per millennium (Buizert et al. 2014). However, the total temperature change during the summer season,
which is largely responsible for controlling ice-sheet melt, was closer to 12°C (Buizert et al. 2018). During the early Holocene,
more muted warming (~3°C over 3 kyr) drove GIS retreat to behind its present-day margin in many sectors (e.g. Bennike &
Weidick, 2001, Larsen et al. 2016, Young et al., 2021). Thus, to capture a reasonable range of warming rates based on
paleoclimate evidence in our ensemble, we subject the ice sheet to an interglacial warming ramp ranging from 1.0 °C kyr$^{-1}$ to
2.0 °C kyr$^{-1}$ in increments of 0.33 °C.

### 2.2.4 Solid-Earth relaxation time

Solid-Earth dynamics influence ice-sheet stability (Austermann et al. 2015) and have changed beneath Greenland as a function
of time (Rogozhina et al. 2016) and potentially in response to fluctuations of the GIS itself (Stevens et al. 2016). A great deal
of uncertainty remains in the details of the solid-Earth beneath the ice sheet (Kappelsberger et al. 2021). Here, we choose to
use two mantle relaxation times (500 and 3,000 years) to represent end-member scenarios, with the former representing hot,
low-viscosity (fast-responding) mantle like that underlying northeast Greenland today (Fahnestock et al. 2001), and the latter
is a standard value for relaxation time that has been calibrated against measurements of glacial-isostatic adjustment (Le Meur
& Huybrechts 1996, Coulon et al. 2021). The strength of this approach within our modelling framework is that it allows us to





quantify where solid-Earth processes are likely critical for understanding sea-level potential without having full knowledge of the true Earth structure beneath Greenland.

**2.2.5 Ice-sheet initialization**

Following its expansion to the continental shelf edge at the end of the Last Glacial Maximum (LGM; 21 ka), rising global $CO_2$ drove the GIS to recede following the LGM and approach its present-day margin (Cuzzone et al. 2019). A warm summer orbit during the HTM (~8ka) led to continued ice-sheet margin retreat inland from its current position (e.g. Young et al. 2021). These two phases of retreat illuminate two distinct ways that Greenland may have deglaciated more fully in the past: either quickly following a glacial period, when the GIS was defined by extensive marine-terminating margins, or after reaching a modern-like "interglacial" state, when much of the ice-sheet is land-terminating and ice-ocean interactions are mostly confined to narrow fjords. To capture both these possibilities, we start our simulations from either an ice sheet that has been run to equilibrium with LGM climate conditions, or a modern ice-sheet. For the latter, we ran a set of simulations with a cold-start (initialized to match present-day ice thickness observations) and a set of simulations that has been spun-up to modern through a glacial cycle (Buizert et al. 2018). To initialize the cold-start modern ice sheet, we used an observational data set of ice extent and thickness (Morlighem et al. 2017). For the spun-up modern ice sheet, we equilibrated the sheet with an LGM climate forcing (Buizert et al. 2018) for 80kyr followed by an evolving climatology for 21kyr (Buizert et al. 2018). The two modern initial ice sheets differ in their geometry near the margin but have a similar ice-volume.

**2.3 Sea-level potential calculation**

To analyse our results, we introduce sea-level potential, defined for every grid-cell (10 km by 10 km) in the model domain as the sea-level equivalent volume of ice lost from the GIS when that grid-cell first becomes ice-free. We define sea level potential as the median value of the histogram depicting the GIS contribution to sea-level from the ensemble when a particular site is ice-free; the uncertainty in this value is defined as the full width of that histogram (Figure 2). The parameter that reduces the spread of 'Greenland contribution to sea-level (m)', or the x-axis of Figure 2b, is the parameter that the GIS responds most sensitively to. We calculate a quantitative sensitivity score by dividing the width of the histogram for a particular subset of the ensemble by the width of the full histogram. When these widths are equal, the sensitivity score is one and the width of the histogram does not change if we consider only a particular value of that parameter. When a parameter spans a smaller range than the full ensemble does, this number is less than one, indicating that knowledge of this parameter reduces the spread of sea level potential. The parameter that each location is most sensitive to is determined by identifying the smallest sensitivity score, and the relative importance of each parameter is determined by the ranking the scores from lowest to highest (Figure 3).



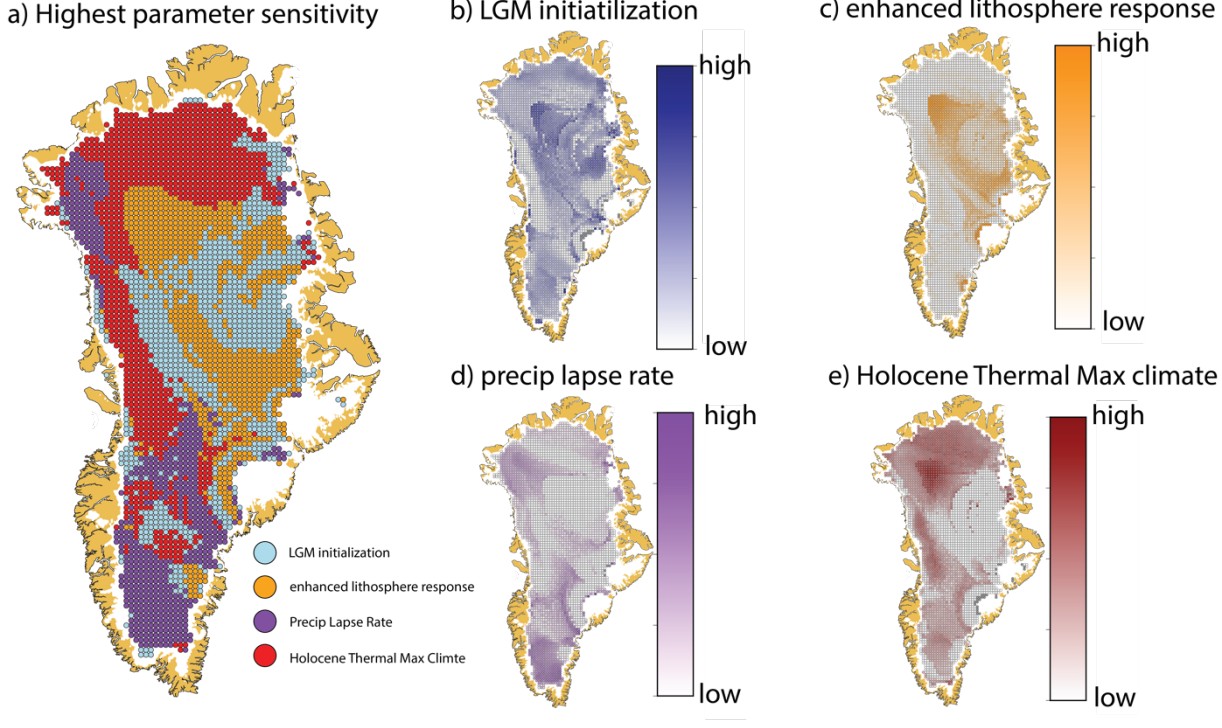

**Figure 3:** Parameter sensitivity test. A) Shows which ensemble parameter exerts the strongest control on the distribution of ice volume estimates when that location first becomes ice free. B) Sensitivity to starting the simulation from Last Glacial Maximum conditions. C) Sensitivity to a reduced response time of the elastic lithosphere relaxing asthenosphere solid-Earth model. D) Sensitivity to neglecting a precipitation lapse rate correction. E) Sensitivity to starting from a climatology from the Holocene Thermal Maximum.

## 3 Results

Sea level potential is generally lower near the ice margins (the first regions to deglaciate in our ensemble) and higher in both Central Greenland and in areas of high topography along the southeast coast (the last regions to deglaciate in our ensemble) (Figure 4b). In southern Greenland, sea level potential increases gradually from the west coast toward the ice divide. In southeast Greenland, areas of high sea-level potential along the margin are resilient to deglaciation, though there is a region between the southern dome and the main dome of the ice-sheet with low sea-level potential. In the north, sea-level potential is generally low near the ice margin and increases towards the ice divide, with a more gradual increase in the northeast and a sharper increase in the northwest. An area of higher sea-level potential extends from the northwest corner of the GIS inward toward central east Greenland, forming a core of high sea-level potential there.

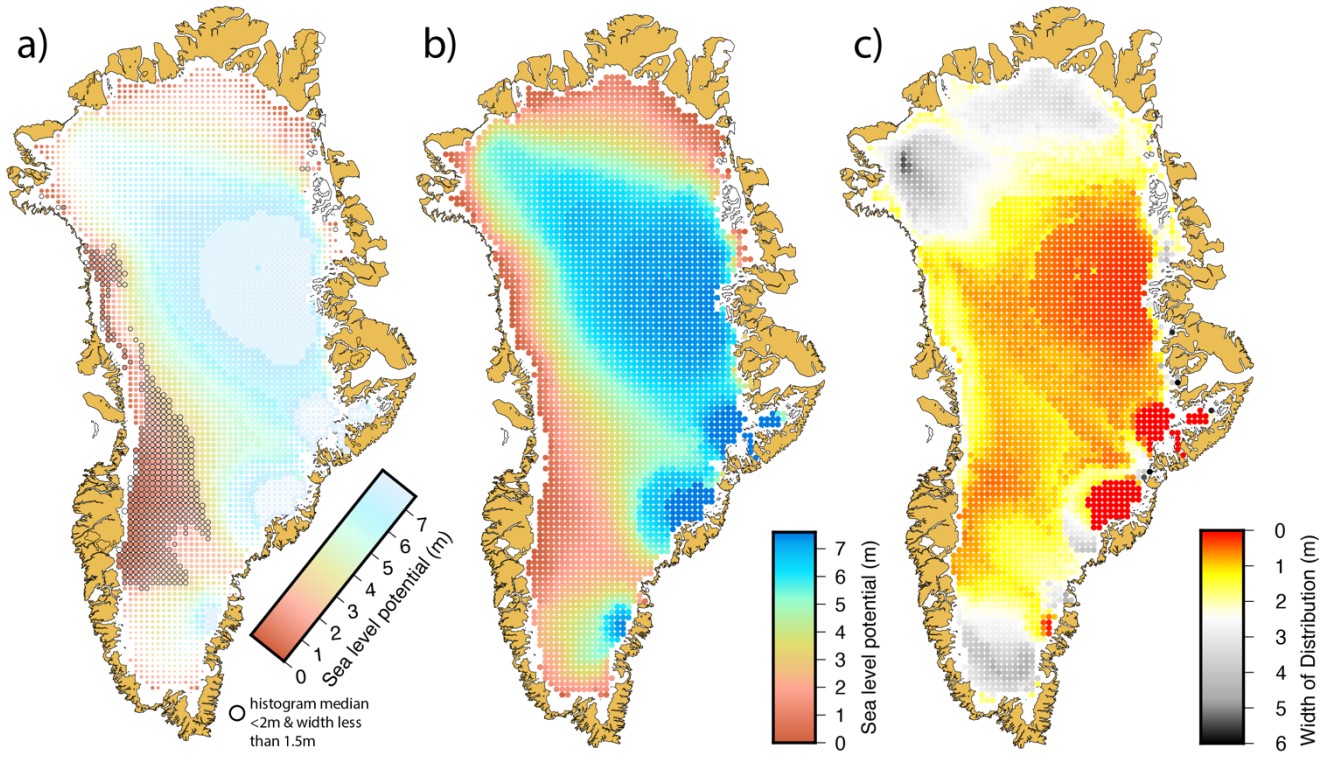


**Figure 4. Greenland's sea level potential. a) Colors indicate sea level potential, defined as the mean amount that Greenland has contributed to global sea level when that grid cell has become ice-free. Size of each dot indicates the uncertainty (quantitatively, the width of the full histogram as in Figure 2b; smaller dots represent a greater uncertainty). Black outline highlights regions where ice-free conditions are associated with median sea-level potential less than 2 meters, and the spread is less than 1.5 meters. Dots are**
**plotted for areas where the ice thickness is greater than 600 meters and sea level potential is greater than zero within a 20 by 20 kilometer area; areas within the modern limit of the ice sheet (underlying white area) that do not meet these criteria are not plotted. b) Sea level potential only (meters sea level equivalent). c) Confidence: Histogram width only (meters sea level equivalent).**

The uncertainty in sea level potential is greatest in North and South Greenland, indicating that ice-free conditions here are
associated with a range of ice-sheet geometries (Figure 4c). The lowest uncertainties are well-correlated with the regions of highest sea-level potential, near where the ice sheet is thickest today and along the southeast coast, where ice caps covering the alpine peaks are the last vestiges of the GIS to melt (Figure 4c). However, there are also relatively low uncertainties throughout West Greenland, which generally decrease towards the ice divide, indicating that the initial stages of deglaciation show a greater variety of ice geometries but as deglaciation progresses these geometries tend to converge. We combine the sea
level potential (median of histogram) and uncertainty (width of histogram) to produce a map that highlights areas that (a) deglaciate when the ice sheet has contributed less than 2 meters to SLR and (b) have an ensemble spread (histogram width) of less than 1.5 meters (Figure 4a). This map reveals regions in Greenland where ice-free conditions are associated with a narrow





band of contributions to sea level (Figure 4a), primarily in west Greenland, but also at select coastal sites in northwest and northeast Greenland (Figure 4a).


We find that initial geometry, aesthenosphere relaxation time, lapse rate for precipitation, and starting climatology play a dominant role in determining sea level potential for different parts of the ice sheet. The starting climatology and precipitation-lapse rate generally play a greater role near the ice-sheet margin, with lithospheric response time and starting ice-sheet geometry playing the dominant role in inland regions (Figure 3a). To identify the most important drivers for each region, we

generated an estimate of the parameter sensitivity for each of our four ensemble parameters (Figure 3b-e). We find that North and West Greenland are most sensitive to the choice of starting climatology, driven by the differences in SMB patterns between the chosen reconstructions for those sectors (Figure 3a). However, broad regions in Northwest and South Greenland are most sensitive to the precipitation-lapse rate scaling. In Central and East Greenland, both the initial configuration (LGM versus modern) and a more responsive solid-Earth are the main factors that drive variance in the ensemble. Across our ensemble,

these regions are consistently the last to deglaciate. Thus, we find that areas near the margin are most sensitive to differences in surface mass balance, whereas inland regions are more sensitive to slower processes like glacial isostatic adjustment that impact ice-sheet evolution over thousands of years. In general, regions that are sensitive to the starting geometry are also sensitive to lithospheric relaxation, and regions sensitive to climatology are also sensitive to precipitation lapse rate. However, there are some exceptions to this trend; for example, in North Greenland climatology and starting geometry are the two

parameters that have the greatest impact on sea level potential there.

## 4 Discussion

### 4.1 Parameters underlying variability in sea-level potential

Incorporating multiple sources of uncertainty in our ensemble design allows us to look in detail at how each parameter impacts the sea-level potential at each site. Areas with a wide spread of sea level potential (e.g. Camp Century; Christ et al. 2023) can

be thought of as places where ice-cover at that site is associated with a wide range of potential ice-sheet geometries; the ice sheet can grow and shrink in different regions while ice remains at that site. Application of this technique to the Camp Century site revealed that ice-free conditions there require a contribution of +1.4 m SLE, although the site can remain ice-covered even when the ice-sheet has contributed +5.6 m SLE (Christ et al. 2023). Uncertainty associated with the sea-level potential could be reduced by adding constraints on simultaneously ice-free conditions at more than one location or by considering a subset

of our parameter space. For example, at Camp Century, simulations that start with an LGM ice sheet require that the ice sheet loses +2.2 meters of sea level equivalent prior to Camp Century deglaciating, and simulations that use a HTM-like climatology require +2.7m before Camp Century deglaciates (Christ et al. 2023). For many sites, the shape of the full histogram suggests that future work to more fully sample the parameter space will be beneficial; nevertheless our end-member approach here




shows that the uncertainty in sea level potential can be greatly reduced in some regions through knowing one of our ensemble
parameters more precisely.

Southwest and North Greenland, where there are broad areas of low sea-level potential that could be accessed by subglacial
sampling, are most sensitive to the spatial climatology pattern (Figure 3a). The inclusion of a precipitation-lapse rate correction
and initializing the simulations with a LGM ice sheet geometry are the dominant parameters in some sub-regions, for instance
in Northwest Greenland and Southwest Greenland. Considering the sensitivities of each individual parameter, the spatial
climatology, precipitation-lapse rate, and LGM initialization all play some role in controlling the ensemble spread in the
regions of Greenland where the first few meters of SLR are likely to be sourced. In contrast, accounting for an enhanced
lithospheric response only impacts the ensemble spread around the most resilient portions of the ice sheet; by the time the ice
margin has reached these areas, Greenland has most likely contributed >4 meters to SLR (Figure 4b). This may reflect a critical
role for solid-Earth processes in dictating the location of the ice-sheet margin in Central Greenland and aligns with a region
that has previously been argued to have a higher geothermal flux and a more viscous mantle (Fahnestock et al. 2001,
Rohogzhina et al. 2014, Stevens et al. 2016). While lithospheric response exerts a dominant control on sea-level potential in
Central Greenland, this source of uncertainty is not likely to impact the regions where the first meter of SLR will come from.

Initializing the model with a LGM ice-sheet geometry has the greatest impact in Central Greenland, where the LGM ice sheet
was thinner than the modern ice sheet due to an arid LGM climate. However, this parameter is of secondary importance for
North and South Greenland and has the least impact in West Greenland. Neglecting a precipitation-lapse rate has the strongest
control on the ensemble in Northwest and South Greenland, where separate ice domes exert a strong control on ice dynamics
(Figure 3d). The dominance of the precipitation-lapse rate illustrates the importance of accounting for changes in precipitation
as temperature and surface elevation changes over peripheral ice-domes during periods of deglaciation (such as Northwest and
South Greenland), as this can increase resilience to the elevation-surface mass balance feedback (Weertman 1961, Edwards et
al. 2014).

The use of a HTM climatology influences deglaciation in North, West, and South Greenland. Holocene melt records are
available in North Greenland, including at NEEM (NEEM Community Members 2013), GISP2 (Alley and Anandakrishnan
1995), and Agassiz ice cap (Koerner et al. 1990). The climate record from NEEM also indicates a greater sensitivity to HTM
conditions, showing an early Holocene mean annual warming of 6ºC, relative to 2ºC at Summit (Lecavalier 2017, Dahl-Jensen
et al. 1998). Our ensemble does not include variations in ocean or indirect sea ice forcing, which likely played a role in past
deglaciation scenarios (Koenig et al. 2014, Irvali et al. 2019). At the fjord scale, ocean warming can have a distinctive impact
on ice-sheet dynamics and thus should be considered in future work (e.g. Straneo et al. 2009, Wood et al. 2021). However,
because the modern GIS is mostly terrestrial and the influence of ocean forcing is often limited to within ~ten ice thicknesses
of even marine-terminating glaciers (Felikson et al. 2017), ocean forcing is expected to have less of an impact on deglaciation





than change in surface climate – although future work to directly test this hypothesis is warranted, in particular for looking at time periods when the GIS was larger than today but smaller than glacial maximum conditions. In addition to playing an important role in the modern-day ablation zone of West Greenland, central northwest Greenland is particularly sensitive to this parameter. This area corresponds to a low-lying part of Greenland's topography and is on the ice divide connecting northwest Greenland with the central dome of the ice sheet. The dominance of the climatology here reflects the important role of HTM-like conditions (enhanced warming in the North and West) for driving deglaciation further once the northwest GIS has disintegrated.

A key control on patterns and rates of deglaciation in regions of low sea-level potential is the applied SMB forcing (e.g. Plach et al. 2018). In our ensemble, the starting SMB fields, particularly the extent of the ablation zone, play an important role in ice-sheet geometry during deglaciation across all scenarios and play a dominant role in sea-level potential for regions that consistently deglaciate before Greenland has contributed its first meter to SLR. Surface mass balance is difficult to accurately reconstruct for past interglacial periods (e.g. Helsen et al. 2016). Our approach circumvents the need for direct reconstruction of SMB for a particular interglacial by considering a range of forcings and identifying the range of sea-level potential associated with the uncertainty in the climate forcing. By including both a pre-industrial and HTM climate forcing, we capture two well-documented modes of interglacial climate in Greenland (e.g. Buizert et al. 2018). However, other modes of surface climate are possible, and may become dominant in the future as boundary conditions and forcings evolve (e.g. Koenig et al. 2014, Sellevold et al. 2021). Our climate forcings come from one reconstruction, and future work to include other work that reconstructs climate using different methods would complement and expand our analysis (e.g. Badgeley et al. 2020). Nevertheless, our results confirm the primacy of correctly predicting the spatial patterns of climate over Greenland (Edwards et al. 2014) for projecting the first meter of future sea-level change, and suggest that selecting sites that have lower uncertainty in their sea-level potential will increase the potency of subglacial observations.

## 4.2 Comparison with other modelling studies

Ice-sheet modelling experiments investigating GIS response to past warmth have resulted in divergent conclusions about ice-sheet stability. Many previous studies found that West Greenland responded most strongly to past interglacial warm periods (e.g. Greve 2005, Robinson et al. 2011, Born and Nisancioglu 2012, Helsen et al. 2015, Sommers et al. 2021). At the same time, other studies have found that North Greenland is also highly sensitive to past interglacial warmth (e.g. Stone et al. 2013). Some studies show both West and North Greenland responding to past warmth simultaneously (Robinson et al. 2011, Born and Nisancioglu 2012, Aschwanden et al. 2019). Our sensitivity-mapping approach allows us to consider how and why these results may differ from other studies that modelled Greenland deglaciation patterns. For example, we find that whether Northern Greenland is an early contributor to SLR is dependent on the choice of a HTM-like climate forcing (Figure 3a). Our approach is distinct because rather than considering one particular warm period, our ensemble encapsulates a range of deglaciation scenarios and treats them as all equally likely. This allows us to overcome the challenges associated with perfectly



simulating a particular time period in favor of identifying the patterns that hold true regardless of the style of deglaciation, and therefore provide useful insight into the uncertain future of the GIS.

### 4.3 Implications for interpreting sea-level records

Our approach complements far-field sea-level records by providing a method to assess where the first few meters of sea-level

rise from Greenland are likely to originate, regardless of the final geometry of the ice sheet at the time of maximum retreat during an interglacial (e.g. Dyer et al. 2021, Barnett et al. 2023). Terrestrial records from West Greenland have revealed this area was particularly sensitive to warming during the HTM (e.g. Larsen et al. 2016, Young et al., 2021) and our results confirm this as a persistent feature of GIS response to warming. Although most far-field sea level records of past warm periods have focused on the maximum sea level attained, our work may provide a framework for thinking about where we may look for

evidence of retreat early in a warm period. Our ensemble identifies SW Greenland as earliest to deglaciate regardless of uncertainty in the climate forcing and other parameters examined here. In part of the parameter space, namely those simulations with a HTM climate forcing and no precipitation lapse rate, north Greenland is also among the first regions to deglaciate.

At present, the ensemble has been designed specifically to demonstrate the concept how subglacial material documenting past

ice-free conditions, in combination with numerical ice-sheet modelling, can provide robust estimates of continent-wide sea level contribution. Due to hysteresis effects (Robinson et al. 2012), sea-level potential may be different for a growing versus shrinking ice sheet; here we focus on the application of sea level potential to presently ice-covered sites as they become deglaciated, but we highlight that future work to examine the hysteresis of sea level potential is warranted and may be useful for understanding what feedbacks and processes are most important for ice-regrowth, which is a critical process for

understanding the full response of the ice sheet to climate change during the Pleistocene (Pico et al. 2018, Dalton et al. 2019). In future work, we plan to incorporate other kinds of constraints, for example the total amount of time that a site was ice free or the need for ice to re-advance over the site as opposed to re-nucleating as a separate ice cap before coalescing with the rest of the ice sheet.

### 5 Conclusions

We present the calculation of sea-level potential as a novel method to constrain past ice-sheet geometry and its associated uncertainty within an ensemble-based ice-sheet modelling framework. Our results reveal regions of the GIS which, when ice-free, are associated with a narrow range of ice-sheet geometries. Future efforts, including the U.S. National Science Foundation-funded GreenDrill (Briner et al. 2022) and European Union Synergy-funded Green2Ice programs to collect samples from beneath the ice-sheet margins and interior, in combination with our results, may provide more robust constraints

on paleo sea level than have been possible with other methods. Moreover, this modelling approach can be used to inform future drilling, as regions we identify represent locations where information about past ice-free conditions can be most directly



translated into information that can inform adaptation efforts. Future work to expand the ensemble of past ice-sheet geometries is required to further develop the usefulness of this method. In particular, we expect that improved knowledge of the past spatial mass balance patterns and relationships between temperature and precipitation change will have the greatest impact on

our results.  Our results reveal distinct spatial patterns of ice-sheet sensitivity to different physical processes, which can provide input to scientific communities working on understanding these processes in space and time; for highly vulnerable regions of GIS such as southwest Greenland, the spatial climatology pattern, treatment of precipitation-lapse rate, and ice-sheet starting geometry influence how the ice-sheet deglaciates. Reducing uncertainty in sea-level projections for both paleo and future scenarios will require efforts to better constrain these parameters. More precise knowledge of e.g. past climate forcing can be

readily incorporated into our experimental design so that sea-level fingerprinting and local sea-level impact predictions are informed by the most relevant sources of paleo-data as Earth's climate continues to warm.

*6 Code Availability.* This work uses the model code described in DeConto et al. (2021)


*7 Data Availability.* Model runs will be archived at the US Arctic Data Science Center and are also available from BAK upon request.

*8 Author Contribution.* Conceptualization: BAK, JMS, RMD; Formal analysis and Methodology: BAK, RMD; Writing –

original draft preparation: BAK; Writing – review & editing: all authors.

*9 Competing Interests.* The authors declare they have no competing interests.



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
