# Peer review of "An ice-sheet modelling framework for leveraging sub-ice drilling to assess sea level potential applied to Greenland"

_EGUsphere, 2024_

## Referee Comment (RC1)

Review of Keisling et al., «**An ice-sheet modelling framework for leveraging sub-ice drilling to assess sea level potential applied to Greenland** »

**General comments**

This study provides an interesting framework in which an ensemble of ice sheet model simulations is used to (1) estimate which regions of the Greenland Ice Sheet are most vulnerable, and (2) assess which of the uncertain forcings (climate, solid-Earth rebound, etc) and boundary conditions (initial ice sheet geometry) dominate the simulated ice sheet retreat uncertainty in particular regions.

The manuscript is very well written, proposes a novel framework, and is an interesting read. I think that the result can be interesting for a wider community than the submitted title (for example) suggest. Overall, the work is well presented, but there are a couple of inconsistencies that should be addressed and/or discussed, see specific and technical comments below.

**Specific Comments** (not in order of significance)

1. Title. This work is relevant also for other researchers than those interesting in sub-ice drilling. The current title also makes the scope too narrow for TC. Why not be bold and rewrite to "An ice-sheet modelling framework to determine vulnerable regions of the Greenland Ice Sheet" or similar.
2. Abstract: Lines (L) 25-27. This sentence is very unclear. Please rewrite. L139-143 provides a very clear and nice summary.
3. Keep in mind that the readers might not be very familiar with all ice core locations and names. For example, please change to: "In Greenland, long-archived basal rock and sediment…" (L71-72).
4. Methods: Section 2 starts with a nice summary and Table, but this leaves the reader with many questions on the specific choices of the parameters. Maybe the reader could be guided if in Table 1 you refer to the specific Sections 2.21-2.2.5 for detailed info on the parameter choices. An additional sentence before L157 "We first explain the sea-level potential, and then give details on the model and simulation set-up." or similar, would also help.
5. Initial climate forcing: Past changes in climate over Greenland were mostly driven by changes in greenhouse gases (e.g. $CO_2$) and changes in insolation. The former causing temperatures in all seasons to increase or decrease, while the latter strongly impacts the seasonal cycle. Are the spatial patterns more important than the seasonal changes (ref: Parameter is called "Spatial climatological pattern" in Table 1; Fig1a showing annual mean (right?) temperatures)? How representative are the early Holocene/Holocene Thermal Max and the PI? Related to this, should they be representative for past interglacials during the entire Pleistocene or for future climate change? Or could they be for both? Is it possible with your modelling framework to discuss/separate the impact of $CO_2$ forcing versus insolation (seasonal impact) forcing? Some more discussion and clarifications regarding the climate forcing is needed.

6. Climate forcing: How do you deal (or not) with the SMB-elevation feedback? Are all ice sheet grid cells always forced by the same initial climate forcing, or is the SMB corrected for the lowering of ice surface elevations during the retreat?

7. L210: TC is read by non-paleo researchers, and PI then does not seem to be the most logical choice to represent "increased atmospheric CO2". Please rewrite to emphasize that this is the case compared to glacial periods.

8. L213: How do you downscale from a 40 km resolution to the 10 km resolution of the ice sheet model? This might not be trivial for SMB.

9. Consistency: The parameters have various naming in Table 1, the figures, and the main text, please make this consistent throughout the manuscript. Holocene Thermal Max or early Holocene? Lithospheric relaxation time, or aesthenosphere, or mantle relaxation times? Modern transient (Table 1) or deglacial spin-up (Fig 1b)? Etc.

10. Precipitation lapse rate: This is notoriously difficult to account for, so I appreciate the effort. However, precipitation also changes spatially due to atmospheric changes (changing climate), and when the shape/surface topography of the ice sheet changes. I assume that this is not represented in your model set-up? Can you include a bit more discussion on this?

11. Run time: Do I understand correctly that all simulations are ran for 10,000 years, and that most of the analyses are done with the final state of the ice sheet (i.e. at year 10,000)? Using a shorter simulation time (interglacials normally do not last 10,000 yrs), or higher rates of warming, would impact how much of Greenland would be deglaciated in these simulations. Would this impact the calculated sea-level potential and uncertainty? Please discuss why you choose 10,000 years and what the impact of a shorter period would be.

12. Solid-Earth: the values from Le Meur & Huybrechts, and Coulon, are for Antarctica, right? (L241-242) Please make this clear in the text.

13. For the reader it makes more sense to first see and read the results related to Fig. 4 (overall results), and then the analyses related to Fig. 3 (specific impacts of parameters). Is it possible to change the order?

14. L439: "robust constraints": this is a big claim. I am not sure if this work can really give robust constraints, but I do see its value in pointing out the vulnerable regions and impacts of uncertain forcings…

15. One last thing: Have you compared your simulated rates of change (in mm SLE/yr or similar) to other studies (e.g., Vasskog et al., 2015; Briner et al., 2020)? This could help constraining the rates of warming or simulation length, and/or give some more general constraints to your work.

**Technical corrections**

1. References: Something seems to have gone wrong with the notation of the references, many commas are missing. It should be (Name et al., year) or (Name and Name, year).

2. Also, sometimes a few references are mentioned, but these are just examples of work. These should include "e.g.,". For example L104 should be (e.g., Helsen et al., 2023, …)

3. Fig. 1B: should the difference for the LGM not be created on the LGM grid, to emphasize the additional ice present outside of the present area?

4. Fig. 2 caption lacks some info: B) Full ensemble is grey. HTM is in orange (not red). C) Blue is land? Other colours indicate number of simulations predicting deglaciation at the location (right?).
5. L214-215: what does this mean "can be volumetrically"? does this sentence miss a word?
6. L233, should "ramp" be "rate"?
7. L 247: omit "following the LGM"
8. L259: Can you state the ice volume values for these 3 initial states?
9. Fig. 4a: very difficult to properly read the size of the dots. I suggest omitting (a), and add the black outlined dots to the map of (currently) Fig 4b.
10. L 313: add "These regions indicate likely regions for the first 1-2 m of ice loss" or similar, just after "less than 1.5 meters (Figure 4a).
11. L399: "potency"? do you mean "impact"?
12. L438: Green2Ice is a "ERC Synergy Grant funded by the European Union"

**References**

Briner, et al., Rate of mass loss from the Greenland Ice Sheet will exceed Holocene values this century, Nature, 586, 70–74, https://doi.org/10.1038/s41586-020-2742-6, 2020.

Vasskog, et al. (2015), The Greenland ice sheet during the last glacial cycle: Current ice loss and contribution to sea-level rise from a palaeoclimatic perspective, Earth-Science Reviews, 150, 45-67, doi:10.1016/j.earscirev.2015.07.006

---

## Author Comment (AC1)

We thank the reviewer for their careful reading and encouraging comments on our manuscript. We have addressed their comments in-line in bold text. Figure captions and line numbers refer to the revised manuscript.

Review of Keisling et al., «An ice-sheet modelling framework for leveraging subice drilling to assess sea level potential applied to Greenland »
General comments
This study provides an interesting framework in which an ensemble of ice sheet model simulations is used to (1) estimate which regions of the Greenland Ice Sheet are most vulnerable, and (2) assess which of the uncertain forcings (climate, solid-Earth rebound, etc) and boundary conditions (initial ice sheet geometry) dominate the simulated ice sheet retreat uncertainty in particular regions.
The manuscript is very well written, proposes a novel framework, and is an interesting read. I think that the result can be interesting for a wider community than the submitted title (for example) suggest. Overall, the work is well presented, but there are a couple of inconsistencies that should be addressed and/or discussed, see specific and technical comments below.

Specific Comments (not in order of significance)
1. Title. This work is relevant also for other researchers than those interesting in sub-ice drilling. The current title also makes the scope too narrow for TC. Why not be bold and rewrite to "An ice-sheet modelling framework to determine vulnerable regions of the Greenland Ice Sheet" or similar.

**Changed to "An ice-sheet modelling framework to determine vulnerable regions of the Greenland Ice Sheet in the past."**

2. Abstract: Lines (L) 25-27. This sentence is very unclear. Please rewrite. L139-143 provides a very clear and nice summary.

**L25–27 have been updated to read "We map the GIS response to warming, in order to (1) estimate of the region(s) of GIS that are most likely to contribute to the first meter of global sea-level change, (2) guide future sub-glacial access efforts that can provide targeted information about the response of the ice sheet to past warming, and (3) contextualize existing and future datasets within a glaciologically coherent, full-geometry framework to establish the minimum GIS contribution to past sea level when a particular location is ice-free." See revised text lines 26–30.**

3. Keep in mind that the readers might not be very familiar with all ice core locations and names. For example, please change to: "In Greenland, long-archived basal rock and sediment…" (L71-72).

**Changed the wording to what the reviewer suggested (see lines 76–77).**

4. Methods: Section 2 starts with a nice summary and Table, but this leaves the reader with many questions on the specific choices of the parameters. Maybe the reader could be guided if in Table 1 you refer to the specific Sections 2.21-2.2.5 for detailed info on the parameter choices. An additional sentence before L157 "We first explain the sea-level potential, and then give details on the model and simulation set-up." or similar, would also help.

**The caption of Table 1 has been updated as the reviewer suggested. We have added a sentence**

**before L157 that reads: "We first provide details on the model and simulation set-up, and then explain our calculation of sea-level potential." The streamlining to keep the definition of sea level potential to section 2.3 was suggested by reviewer #2.**

5. Initial climate forcing: Past changes in climate over Greenland were mostly driven by changes in greenhouse gases (e.g. CO2) and changes in insolation. The former causing temperatures in all seasons to increase or decrease, while the latter strongly impacts the seasonal cycle.

**This is correct, and this is indeed reflected in the forcing that we use, because the HTM spatial climatology has relatively warmer summers particularly at higher latitudes (Figure 1A).**

Are the spatial patterns more important than the seasonal changes (ref: Parameter is called "Spatial climatological pattern" in Table 1; Fig1. a showing annual mean (right?) temperatures)?

**We have edited the figure caption to clarify that these are indeed mean annual temperatures shown. We have also updated the figure to show the difference in summer (July) and winter (DJF) temperatures separately to enrich the discussion about the differences between these climatologies and the ways that they can represent different end-members of Pleistocene climate forcing. Indeed, the Holocene Thermal Maximum climatology is broadly characterized by warmer summers, particularly in the north and west, but colder winters (revised Fig 1). We can see in more detail the differences in these climatologies if we look at the average monthly temperature averaged for central Greenland (here defined as latitude 75.5–80.0ºN, 41.5–61.5ºW):**

[Figure]

**The HTM is warmer during the summer months, and colder during the winter months, meaning the amplitude of the seasonal signal is enhanced (33.8ºC compared to 29.4ºC for preindustrial). Overall, the mean annual temperature is actually lower during the HTM (-24ºC) compared to the preindustrial (–22.8ºC). This is because the HTM climate changes are responding to changes in insolation but under slightly lower $CO_2$ concentrations.**

**It is difficult to disentangle or separate the influences of seasonality versus spatial pattern in the reconstructions using our current methodology, because the reconstruction we used explicitly accounts for changes in both insolation and $CO_2$ simultaneously. For summer melting, which is a key process for deglaciation, seasonality is likely playing a dominant role because the HTM climatology is several degrees warmer than the PI climatology during the summer months, as shown in the figure above. We agree that seasonality is an important process that could be considered in detail in future work, using the framework we describe here (e.g. by analyzing an ensemble that includes different treatments of seasonality).**

How representative are the early Holocene/Holocene Thermal Max and the PI? Related to this, should they be representative for past interglacials during the entire Pleistocene or for future climate change? Or could they be for both? Is it possible with your modelling framework to discuss/separate the impact of CO2 forcing versus insolation (seasonal impact) forcing? Some more discussion and clarifications regarding the climate forcing is needed.

**To continue on the above response, these reconstructions consider both insolation and $CO_2$ forcings, so they only represent two scenarios – a "high $CO_2$/low insolation" scenario (preindustrial) and a "low $CO_2$/high insolation" scenario (HTM). In this way, they are representative of two different modes of interglacial warmth, and the preindustrial climatology captures the modern-day seasonality well. But, but they do explicity capture a high-$CO_2$/high-insolation scenario, nor do they capture the potential for a different kind of changing seasonality moving into the future, as has been suggested by some future modelling studies, because these scenarios are not captured in the last 21kyr. We have added an additional discussion on this point in the revised manuscript section 2.2.1, and we thank the reviewer for the inspiration to discuss these interesting points in greater detail in lines 227–231: These timeslices thus represent two end-members: lower $CO_2$/high insolation (early Holocene) and higher $CO_2$/lower insolation (preindustrial). In this way, they are representative of two known modes of interglacial warmth, and capture both the spatial and seasonal patterns associated with them. However, they do not capture different spatial/seasonal patterns that might be associated with climates warmer than modern, and we assume both spatial and seasonal patterns stay fixed as we conduct the interglacial warming experiments.**

6. Climate forcing: How do you deal (or not) with the SMB-elevation feedback? Are all ice sheet grid cells always forced by the same initial climate forcing, or is the SMB corrected for the lowering of ice surface elevations during the retreat?

**We use a linear temperature lapse-rate correction of 5.0ºC $km^{-1}$ to downscale the 40 $km^2$ climate forcing to the 10 $km^2$ ice-sheet grid, and to dynamically adjust the ice-sheet surface temperature as the ice geometry evolves (e.g. height-mass balance feedback, Weertman 1961). This sentence has been added to section 2.1 (see lines 205–208) in the revised manuscript.**

7. L210: TC is read by non-paleo researchers, and PI then does not seem to be the most logical choice to represent "increased atmospheric CO2". Please rewrite to emphasize that this is the case compared to glacial periods.

**Added "relative to glacial periods" (line 226).**

8. L213: How do you downscale from a 40 km resolution to the 10 km resolution of the ice sheet model? This might not be trivial for SMB.

**Downscaling from the 40km resolution climate reconstruction to the 10km ice-sheet model grid is done using a temperature lapse rate correction of 5.0ºC km$^{-1}$. Melt is then calculated using a positive degree day scheme that applies 5 mm of melt per degree day. Precipitation is calculated assuming no change in precipitation as a function of temperature or a precipitation lapse rate of 2% ºC$^{-1}$. This has been clarified in Section 2.1 of the revised manuscript (see lines 205–208).**

9. Consistency: The parameters have various naming in Table 1, the figures, and the main text, please make this consistent throughout the manuscript. Holocene Thermal Max or early Holocene? Lithospheric relaxation time, or aesthenosphere, or mantle relaxation times? Modern transient (Table 1) or deglacial spin-up (Fig 1b)? Etc.

**We thank the reviewer for catching this inconsistency. We have revised the manuscript to make these consistent throughout, so that the language in the table is the same as that used through the rest of the manuscript.**

10. Precipitation lapse rate: This is notoriously difficult to account for, so I appreciate the effort. However, precipitation also changes spatially due to atmospheric changes (changing climate), and when the shape/surface topography of the ice sheet changes. I assume that this is not represented in your model set-up? Can you include a bit more discussion on this?

**This is correct. Our treatment of precipitation lapse rate only accounts for changes in precipitation that are directly correlated with changes in temperature. This is a first-order correction that is consistent with the broad-scale precipitation patterns known from glacial-interglacial temperature changes (e.g. Alley et al. 1993). However, our model set-up does not capture changes in precipitation that would be driven by other changes in boundary conditions, e.g. different moisture pathways due to changes in sea-ice (e.g. Koenig et al. 2015). Due to the difficulty in reconstructing past changes in precipitation, we leave a more complex treatment of potential precipitation changes to future work, but note that our results can guide understanding of which parts of the ice-sheet are most sensitive to changing precipitation during warm climates (our Figure 4). We have expanded the discussion on this topic in lines 412-415 to read: In particular, although the precipitation lapse rate we apply reveals regions where changes in precipitation strongly impact deglaciation, changes in precipitation that are not associated with temperature changes (e.g. changes in moisture pathways associated with changing sea-ice cover; Koenig et al. 2015) are not captured by our ensemble set-up.**

11. Run time: Do I understand correctly that all simulations are ran for 10,000 years, and that most of the analyses are done with the final state of the ice sheet (i.e. at year 10,000)? Using a shorter simulation time (interglacials normally do not last 10,000 yrs), or higher rates of warming, would impact how much of Greenland would be deglaciated in these simulations. Would this impact the calculated sea-level potential and uncertainty? Please discuss why you choose 10,000 years and what the impact of a shorter period would be.

**Yes, each simulation runs for 10,000 years. However, our analyses are not done with the final state of the ice-sheet, but rather the time that a particular location becomes ice-free within a 10,000 year warming period. We use four rates of interglacial warming (Table 1) to capture different rates of warming. We chose a 10,000 year warming duration because this is a common duration for interglacial periods (reference). Even with 10,000 years of warming, some ensemble members do not completely deglaciate (see our Figure 1). With a shorter duration of warming, we expect that more ensemble members would not fully deglaciate, and some regions would remain ice-covered for**

**the duration of the experiment (imagine Figure 1, but where warming stopped at 5,000 years, such that some ensemble members still had thick ice-cover). Sea-level potential could still be used in this case, but the histogram we use to produce sea-level potential would not be fully populated, which could alter the median and spread, because we would only be sampling the subset of simulations that deglaciate most quickly. We choose 10,000 years as a representative interglacial length, but acknowledge that the sea-level potential for a site could be slightly different for an interglacial of 1,000 years or 30,000 years duration. We have added the following text to section 2.2.3 to clarify this: We choose to apply this rate of warming for 10,000 years as a representative Pleistocene interglacial length, and note that the sea-level potential for a given site may be different for an interglacial of a different duration. For a shorter-duration interglacial, the subset of ensemble members that deglaciate most quickly could be analysed. For a longer-duration interglacial, the simulations could be run further forward with continued warming.**

12. Solid-Earth: the values from Le Meur & Huybrechts, and Coulon, are for Antarctica, right? (L241-242) Please make this clear in the text.

**This has been clarified in the revised manuscript, line 267.**

13. For the reader it makes more sense to first see and read the results related to Fig. 4 (overall results), and then the analyses related to Fig. 3 (specific impacts of parameters). Is it possible to change the order?

**The order of these figures has been changed as suggested by the reviewer, and calls to these figures changed throughout the manuscript.**

14. L439: "robust constraints": this is a big claim. I am not sure if this work can really give robust constraints, but I do see its value in pointing out the vulnerable regions and impacts of uncertain forcings…

**The word "robust" has been removed. The revised manuscript now reads (lines 477–480): Future programs to collect samples from beneath the ice-sheet margins and interior, including the U.S. National Science Foundation-funded GreenDrill (Briner et al. 2022), and Green2Ice, an ERC Synergy Grant funded by the European Union, in combination with our results, may provide novel constraints on paleo sea level contributions from the GIS.**

15. One last thing: Have you compared your simulated rates of change (in mm SLE/yr or similar) to other studies (e.g., Vasskog et al., 2015; Briner et al., 2020)? This could help constraining the rates of warming or simulation length, and/or give some more general constraints to your work.

**Although in this manuscript we do not focus on rates, over all of our simulations the rates of sea-level change are broadly similar to those seen during the Holocene and predicted for the future (Vasskog et al. 2015, Briner et al. 2020):**

[Figure]

**Here we have compared the rates of ice-mass loss during our deglaciation runs with both Vasskog et al. (2015) and Briner et al. (2020). We have calculated the rate of sea-level rise in 100-year bins for every ensemble member and made a histogram of results. In the upper two panels, we have adjusted the x-axis of our histogram to match the limits from the other two publications as best as possible. In the lower panel, the full histogram of our results is plotted, and smaller versions of the two other studies are plotted above for comparison. The solid blue bars are for the data binned in the units of $10^3$ Gt century$^{-1}$, whereas the transparent bars correspond to the same data binned in units of mm yr$^{-1}$. We find that the rates of sea-level change predicted by our ensemble are similar to those predicted for future warming scenarios (red dot from Briner et al. (2020) and colored boxes from Vasskog et al. (2015)) and those experienced by the GIS during the deglaciation and Holocene (red line from Vasskog et al. (2015) and gray histogram from Briner et al. (2020)).**

Technical corrections
1. References: Something seems to have gone wrong with the notation of the
references, many commas are missing. It should be (Name et al., year) or (Name
and Name, year).

**Thank you for pointing this out. We have re-exported the references from Zotero in the Cryosphere format and double-checked that the in-text citations match the reference section.**

2. Also, sometimes a few references are mentioned, but these are just examples of
work. These should include "e.g.,". For example L104 should be (e.g., Helsen et al.,
2023, ...)

**This has been fixed (line 98).**

3. Fig. 1B: should the difference for the LGM not be created on the LGM grid, to
emphasize the additional ice present outside of the present area?

**Figure 1 has been updated to reflect this change. Now each panel shows the difference with BedMachine on the model grid, such that the model initialized for LGM conditions has a footprint that extends beyond the BedMachine dataset.**

4. Fig. 2 caption lacks some info: B) Full ensemble is grey. HTM is in orange (not red).
C) Blue is land? Other colours indicate number of simulations predicting deglaciation
at the location (right?).

**The Figure 2 caption has been adjusted to contain the additional information and corrections that the reviewer pointed out.**

5. L214-215: what does this mean "can be volumetrically"? does this sentence miss a
word?

**Thank you for pointing this out – the sentence was missing the word "offset" and has been adjusted to reflect this (line 235).**

6. L233, should "ramp" be "rate"?

**Changed (line 255).**

7. L 247: omit "following the LGM"

**Changed (line 271–272).**

8. L259: Can you state the ice volume values for these 3 initial states?

**Added the ice volume for each ice-sheet state to the revised Figure 1.**

9. Fig. 4a: very difficult to properly read the size of the dots. I suggest omitting (a), and add the black outlined dots to the map of (currently) Fig 4b.

**This has been changed following the reviewer's suggestion.**

10. L 313: add "These regions indicate likely regions for the first 1-2 m of ice loss" or similar, just after "less than 1.5 meters (Figure 4a).

**Added: "**These regions indicate likely regions for the first 1–2 m of ice loss in the past."

11. L399: "potency"? do you mean "impact"?

**Changed (line 435).**

12. L438: Green2Ice is a "ERC Synergy Grant funded by the European Union"

**Changed lines 478–479 to reflect the language suggested by the reviewer.**

**References**

Alley, R. B., Meese, D. A., Shuman, C. A., Gow, A. J., Taylor, K. C., Grootes, P. M., White, J. W. C., Ram, M., Waddington, E. D., Mayewski, P. A., and Zielinski, G. A.: Abrupt increase in Greenland snow accumulation at the end of the Younger Dryas event, Nature, 362, 527, 1993.

Briner, J. P., Cuzzone, J. K., Badgeley, J. A., Young, N. E., Steig, E. J., Morlighem, M., Schlegel, N.-J., Hakim, G. J., Schaefer, J. M., Johnson, J. V., Lesnek, A. J., Thomas, E. K., Allan, E., Bennike, O., Cluett, A. A., Csatho, B., de Vernal, A., Downs, J., Larour, E., and Nowicki, S.: Rate of mass loss from the Greenland Ice Sheet will exceed Holocene values this century, Nature, 586, 70–74, https://doi.org/10.1038/s41586-020-2742-6, 2020.

Koenig, S. J., DeConto, R. M., and Pollard, D.: Impact of reduced Arctic sea ice on Greenland ice sheet variability in a warmer than present climate, Geophys. Res. Lett., 41, 3933–3942, https://doi.org/10.1002/2014GL059770, 2014.

Vasskog, K., Langebroek, P. M., Andrews, J. T., Nilsen, J. E. Ø., and Nesje, A.: The Greenland Ice Sheet during the last glacial cycle: Current ice loss and contribution to sea-level rise from a palaeoclimatic perspective, Earth-Science Reviews, 150, 45–67, https://doi.org/10.1016/j.earscirev.2015.07.006, 2015.

---

## Author Comment (AC2)

We thank the reviewer for their careful reading and encouraging comments on our manuscript. We have addressed their comments in-line in bold text. Figure captions and line numbers refer to the revised manuscript.

REVIEW 2

Summary

The authors present a new diagnostic they call "sea-level potential" that represents the amount of mass lost from the ice sheet when a particular location first becomes free of ice. The underlying idea is that the combination with sub-ice drilling at specific locations could constrain past sea-level contributions from the ice sheet if recovered basal material allows identification and dating of ice-free conditions some time in the past.

**We are glad that the underlying idea came across clearly to the reviewer.**

In my view, the schematic forcing used renders the presented results as a proof-of-concept rather than an actually usable framework. If it could be applied in practice hinges on large uncertainties in climate forcing in the past, which remain to be resolved. In any case, uncertainties would need to be fully accounted for and/or shortcomings better acknowledged. I suggest mayor revisions are needed to resolve the conceptional problems and reframe the paper.

**We appreciate the reviewer's feedback on this point. We agree that our framework is intended as a proof-of-concept, because we have made specific assumptions about the model set-up and ensemble design to look at shared characteristics of deglaciation scenarios driven by forcings that represent Pleistocene interglacials, but are not particular to one time period. We disagree that this renders the framework "unusable." However, it is important that the reader understands the assumptions and choices we made in opting for a generalized schematic approach to past interglacials, and interprets our results in that context. We have clarified the text in lines 355–358 to encourage that the framework be used in this way, along with being transparent about what the reviewer has identified as shortcomings throughout the manuscript e.g. lines 255, 431–433.**

**We agree that the usefulness of the framework does depend on uncertainties in climate forcing in the past and have identified that as a key next step for improving and using the framework moving forward (see lines 489–490).**

General comments

While constraining ice sheet contributions to sea level in the past could in principle give some indication for future behaviour, I believe the presented method and experiments are not suited to give direct information about the future. The climatic forcing for the future would be different from what is prescribed here and consequently would be the spatial pattern of retreat. I would suggest to refrain from making statements about the future (e.g. l48, l398, l412, l415, l442) based on the presented method.

**We appreciate this feedback. Whether the past is representative of the future is the subject of much debate and disagreement. On one hand, the ice sheet is melting more quickly today than it has during most of the Holocene (Briner et al. 2020), and this historically unprecedented behavior is projected to continue (Aschwanden et al. 2019). On the other hand, the large changes in sea level predicted for the coming centuries only have equivalents in the past (e.g. Dutton et al. 2015). This is an important piece of the motivation for our work. There are many ongoing efforts to better**

**understand how the ice sheet responded to climate change in the past in order to provide context for ongoing changes and potentially inform projections for the future. But, we appreciate that our manuscript does not directly pertain to the future. We have adjusted the text to reduce the emphasis on the future, including in the lines mentioned by the reviewer, while remaining true to the fundamental motivation that the past might hold insights to the future even if it does not predict it. See, especially, revised lines 456–457 and 459–461 where we have re-oriented the text that was describing the relevance of the current work for the future.**

A similar problem arises for specific periods of the past, e.g. the last interglacial. Examples of ice sheet modelling results show a very large range of possible retreat scenarios, largely explained by uncertainties in the climatic boundary conditions, but not only (e.g. Plach et al., 2018). Going further back in time does not improve the constraints either. It is not clear to me how this large uncertainty somehow disappears when using the forcing applied in the presented work. I strongly reject the notion that the forcing and therefore the retreat is generic and independent of a specific deglaciation (e.g. l390 and l411).

**We appreciate the reviewer's comments. The purpose of the schematic forcing is not to make the boundary conditions perfect, but to capture major modes of deglacial climate forcing and understand the shared characteristics of ice-sheet response to those forcings given uncertainty in the initial state, solid Earth properties, and climate-ice sheet feedbacks. We do not mean to imply that the retreat is generic – indeed, every deglaciation is probably different. Instead, we mean to identify common responses among deglaciation scenarios across a range of forcings that does not perfectly represent past interglacials but captures their major characteristics (e.g. driven by insolation vs. $CO_2$, warming at different rates). We have tried to make this clearer in the revised manuscript, for example in lines 227–231.**

The results are obtained from a schematic, continuously increasing forcing, which promotes continuous retreat. How does the method cope with past deglaciations that include periods of intermediate cooling and re-advance? I believe this possibility needs to be discussed as it means a non-uniqueness for the mapping of sea-level contribution to location.

**It's true that it promotes continuous retreat but it does not lead to total retreat in all scenarios, and equilibria are reached. We acknowledge that the focus of this paper is on simple deglaciation scenarios, as a way to understand what kinds of retreat behavior are shared among them. A schematic approach is used because, as mentioned by the reviewer, uncertainties in past climate forcing are large. We agree that it would be worthwhile to look in more detail at deglaciations that are interrupted by cooling and readvance, and our framework would be flexible to incorporate this. We have better framed this work as a proof of concept that demonstrates the benefits of this framework while also highlighting that the results are specific to the particular ensemble (lines 355–358). We have also discussed the importance of reglaciation or glacial advance processes in lines 472–478.**

The proposed "sea-level potential" diagnostic maps ice free conditions around the ice sheet to its global sea-level contribution. In some sense this is similar to "time of retreat", except that time then maps non-linearly to sea-level contribution. The declared advantage of the presented approach is that the time dependence is collapsed and any ice sheet trajectory can be mapped onto the same potential axis. My concern is that this would only work if the retreat was independent of the specific climate forcing, which is clearly not the case. Considering this, what is the advantage of conflating different uncertainties by determining "sea-level when ice-free" rather than finding time of retreat (with uncertainty) and then mapping to sea-level?

**The advantage is that we are interested not in a specific time but in patterns that can reveal that shared vulnerabilities across different realizations of a deglaciation. This is useful for the present context, because this work is motivated by the increasing prevalence of studies where subglacial material is being leveraged to learn about past ice-free conditions without knowing the precise timing of when those conditions occurred. We propose this framework as a different and complementary way to think about model-data integration.**

I can see that for one specific ice sheet trajectory (and ignoring the non-uniqueness above for now) the method could indeed be used to identify relevant drilling sites or argue for certain sea-level constraints in the past. However, the uncertainty of climate and ice sheet model reconstructions for the past (e.g. the last interglacial) remains so large that any practical application is seriously hampered.

**We appreciate and share the frustration that uncertainty in past climate forcing is very large, which greatly hinders our collective ability to understand what exactly drove the Greenland Ice Sheet to be dramatically smaller than it is today during past interglacials. We have conducted this study within that context, and tried to devise a methodology that is general enough to capture the character of different kinds of past interglacial forcing (e.g. warming occurring at different rates, driven more by $CO_2$ or insolation forcing, with or without additional precipitation) so that we can better interpret subglacial observations to say something about the potential ice-sheet geometries they encode. We see this as a very practical endeavor because these samples exist, and are being used to try to understand past ice-sheet contributions to sea level, but they are often not yet precisely dated and the forcing is unknown. Our work addresses the need to use glaciological models with an accounting for some uncertainties in order to make sense of these observations.**

If this method should indeed serve as the basis for determining best locations of multi-million dollar drill campaigns a full understanding of the uncertainties would be needed, rather than trying to minimise them.

**See above – we have not argued that this should be the sole basis for determining the best locations for multi million dollar drill campaigns, which are constrained by a wide variety of factors; however, we do argue that this model can be synergistic with other considerations when seeking to address particular hypotheses that require obtaining new subglacial materials or analyzing long-archived subglacial materials. We did not mean to imply that we are trying to minimize uncertainties – indeed the motivation for this work is better understanding the range of ice-sheet geometries that can be consistent with ice-free conditions at a particular location. We have updated the text (see lines 431–435) to better reflect this.**

A discussion and (where possible) quantification of the full range of uncertainties entering the results should be added. Another alternative for this paper could be to present the approach as a proof-of-concept and acknowledge limitations better.

**The reviewer has already pointed out that the uncertainties in past climate forcing in particular are substantial, and our results, and their associated uncertainties, are indeed tied to the assumptions we have made in the way we designed the ensemble. We have clarified why we designed the ensemble the way that we did (see lines 125–130), better acknowledged the limitations of the present study design (see lines 355–358), and communicate that we indeed see this paper as a proof-of-concept that can be improved as we gain better constraints on presently uncertain boundary conditions, e.g. past climate forcing (lines 380–381, 434–436, 479–481).**

The idea to determine where the next meter of SLR comes from occurs at several places in the

manuscript. I would argue that this is a flawed concept and it is not possible to answer this question with the proposed method.

**We appreciate the reviewer's argument and have reframed this question as "where did the first meter of sea level come from when the GIS was smaller in the past?" wherever logical throughout the manuscript. We still mention the utility of studying the past to better understand the future (e.g. old line 48, new line 50–52), which we maintain is an important and worthwhile scientific approach, but have reworded several sections to avoid any confusion that we are directly making future projections with the modelling approach documented here.**

Most importantly, ice loss from the GrIS may happen everywhere along the margin, whether or not a region becomes ice free. At a particular moment during the deglaciation, mass loss has not only happened in all the places that have become ice free until then, but also possibly everywhere else. Remember, ice is flowing! In other words, mass loss does not happen by removing entire columns of ice. This framing reveals a flawed understanding of the physical system. In addition, problems with specific future climate forcing also apply here.

**That ice loss (and potentially gain) is occurring everywhere is included in our simulations – this is indeed one of the main factors controlling the ensemble spread of sea level potential, and sites where this matters the most have a greater spread – particularly those sites where we find the sea level potential most sensitive to the spatial climatology or precipitation-lapse rate correction (see Figure 4d,e). We apologize that this did not come across clearly to the reviewer and have adjusted the text to try and make this clearer (see lines 239–242).**

Specific comments

Page: 1

l16. There is no direct link between ice sheet response to past changes and future warming. This should be reflected in the sentence.

**Changed wording to say "Estimating the volume of Greenland ice that was lost during past warm periods can support efforts to constrain the ice sheet's response to future warming" (line 16-17).**

l18. "direct information"?

I seem to understand that exposure dating so far has not given these tight constraints.

**Here we mean "direct" in the sense that it is unambiguous that the ice sheet was once absent from its present location, even if the timing is not perfectly constrained. Because exposure dating indeed has reshaped our understanding of ice-sheet absence/presence, we have chosen to keep this language. We do not mean to overstate the potential for such measurements to unambiguously constrain ice-sheet history, and we have adjusted our discussion in lines 78–82 to be more clear about what we mean by "direct":**

**Cosmogenic-nuclide dating (Schaefer et al. 2016), multi-proxy analysis (Christ et al. 2021), and optically stimulated luminescence dating (Christ et al. 2023) in these archives provide direct evidence that the ice sheet was absent from the site within a particular window of time; although these methods can still be ambiguous about the exactly timing of past ice-free conditions, as these methods continue to develop and more samples become available, these windows are narrowing.**

**However, we have chosen to keep the present language in the abstract because these measurements do indeed provide direct information about ice-free conditions.**

l19. "sea-level potential"

Is often used to refer to the total sea-level contribution of an ice sheet if entirely melted. Consider a different term.

**We considered other formulations but we find that the current wording is the clearest and most concise. We have, in response to the reviewers other comments, been more specific in our particular definition of sea-level potential. We note that in other literature, the term "total sea level potential" is used to refer to the total amount of sea-level equivalent in an ice sheet (e.g. Morlighem et al. 2017) and "sea level potential" has been used to estimate sea-level equivalent for individual catchments of an ice sheet (e.g. Greenbaum et al. 2015, Pelle et al. 2024). The latter usage is in line with our use of the term, which we have simply generalized to be defined everywhere rather than for each basin. It is not a particularly overused term in general (8 results in the cryosphere including this manuscript) so we hope that readers will consider our definition and engage with our results accordingly.**

l20. "the amount the GIS has contributed to sea level when a particular location in Greenland is ice-free"

See later for redefinition within an ensemble. Maybe this should be updated already here.

**We have revised the abstract text in this section to be more clear that sea-level potential is defined for a particular location and within an ensemble: "Here, we provide a framework for assessing sea-level potential, which we define within an ensemble of ice-sheet model simulations as the amount the GIS has contributed to sea level when a particular location in Greenland is ice-free." (Line 20–22).**

l25. "Our framework allows us to quantify the local and regional uncertainty in sea-level potential"

**This sentence has been changed to be more clear in response to comments from another reviewer. It now reads, "We map the GIS response to warming, in order to (1) estimate of the region(s) of GIS that are most likely to contribute to the first meter of global sea-level change, (2) guide future sub-glacial access efforts that can provide targeted information about the response of the ice sheet to past warming, and (3) contextualize existing and future datasets within a glaciologically coherent, full-geometry framework to establish the minimum GIS contribution to past sea level when a particular location is ice-free."**

Page: 2

l48. "the questions of when, at what rate, and from where will the next meter of global SLR"

Wrong question. See general comment.

**See our response to general comment. The purpose of this analysis and manuscript is not to project future sea level change, but to provide a framework that can aid in the interpretation of information about past ice-sheet extent in a way that can use observations from a single location to infer whole ice-sheet characteristics, understand the uncertainty in that metric, and investigate what drives it. In other parts of the manuscript (e.g. old lines 412, 415) we have removed references**

**to the future in order to avoid confusing the reader. Here we have chosen to keep this question because this is the Introduction and this is indeed a central question that motivates not only this work but many studies of past ice-sheet responses to climate change.**

l45. "the ice sheet remained extensive and continuous across the island despite >8ºC of warming"

Note that finding interglacial ice alone could not have possibly constrained that. The conclusion was drawn based on modelling experiments not referenced in the paper (Helsen et al. 2013).

**The particular statement from that reference we wanted to reference is:**

**"Even with minimum ice thickness of only about 10% less than the present ice thickness at the NEEM site, as reported here, substantial melting can cause significant reduction of ice thickness near the margins; this in turn reduces the volume of the Greenland ice sheet. Although the documentation of ice thickness at one location on the Greenland ice sheet cannot constrain the overall ice-sheet changes during the last interglacial period, the NEEM data can only be reconciled with Greenland ice-sheet simulations[30] that point to a modest contribution (2 m) to the observed 4–8 m Eemian sea level high stand[44,45]." (NEEM Community Members, 2013).**

**The revised text has been updated to more carefully reflect that statement, and the additional reference suggested by the reviewer has been added for additional context (Lines 57–61):**

**Analysis of interglacial ice preserved at the base of the NEEM ice core in Northwest Greenland (Figure 1) was interpreted to suggest that during the Eemian (Marine Isotope Stage (MIS) 5e, 125 thousand years ago (ka)) the ice sheet contributed only modestly to global sea-level despite >8ºC of warming at that site, with most the ice-loss at the margin (NEEM Community Members 2013); this is consistent with coupled climate-ice-sheet modelling under MIS 5e boundary conditions (Helsen et al. 2013).**

l61. "more-or-less its present configuration for the duration of the Pleistocene"

I am confused about this sentence. The GrIS during MIS11 was likely substantially smaller. Even with Dye-3 intact, one could imagine large retreat.

**We agree and indeed one of the things we are trying to do with this paper is put some quantitative possibilities behind that imagination – e.g. what geometries are possible when different locations are ice-free? These results have however been historically interpreted to suggest that the ice sheet has been stable during the Pleistocene. We have adjusted the revised manuscript to be less confusing: "These results, which argue the GIS did not completely melt during the Pleistocene, are unsurprising in the context of previous modelling results…" (Line 66–71).**

Page: 3

l65. "short enough magnitude and/or duration (e.g. < 1ka) cause changes that are broadly reversible"

Not sure these are still arguments for a GrIS more or less the present size during the entire Pleistocene, but just to clarify: MIS11 lasted for how long? And reversible does not exclude a strong retreat!

**Thank you for these comments. We have adjusted the language in the previous sentence so that "more or less the present size," which we agree is ambiguous and imprecise, is no longer included.**

**We think this makes the transition to these next sentences more clear, and they are intended to articulate other arguments for work that has demonstrated the possibility of a "stable" ice sheet in response to Pleistocene climate forcings.**

**Interglacial duration is difficult to define, (PAGES Working Group, 2015); depending on the definition applied, MIS11 lasted 20–40 kyr. In Greenland, peak interglacial climate as defined by sea surface temperatures in the North Atlantic (Cluett et al. 2021) and terrestrial pollen likely sourced from Greenland (de Vernal et al. 2008) indicate elevated temperatures for ~20kyr (see synthesis in Christ et al. 2023 Figure 1); past modelling approaches have considered a duration of 12.8 to 17.5 kyr for above-freezing local climate anomalies (Robinson et al. 2017). Here, rather than trying to capture a particular interglacial, we use a general approach that is time agnostic to understand the relationship between ice-free conditions at particular locations and the broader geometry of the ice sheet; for this reason, we have assumed a 10 kyr interglacial duration, because this more representative of a general interglacial period (the mean and median duration for Pleistocene interglacials is 9.0 and 7.8 kyr, respectively (threshold method, Table 2, PAGES Working Group (2015)).**

**We agree that reversible does not exclude a strong retreat, and indeed our results show the potential for strong retreat under some Pleistocene interglacial climate conditions.**

l75. Again, not sure 'direct' is the right word here.

**The idea that we are trying to convey is that cosmogenic nuclide concentrations in a subglacial rock or sediment sample provide unambiguous evidence that the material was ice-free within a defined time window. We have replaced 'direct' with 'in situ' to better adhere to this idea.**

l96. "not tied to a particular interglacial"

I don't think it is possible to do that. See also general comment.

**Building here on our response to the general comment: The advantage of our approach is that we are interested not in a specific time but in patterns that can reveal that shared vulnerabilities across different realizations of a deglaciation. This is useful for the present context, because this work is motivated by the increasing prevalence of studies where subglacial material is being leveraged to learn about past ice-free conditions without knowing the precise timing of when those conditions occurred. We propose this framework as a different and complementary way to think about model-data integration. We appreciate and share the frustration that uncertainty in past climate forcing is very large, which greatly hinders our collective ability to understand what exactly drove the Greenland Ice Sheet to be dramatically smaller than it is today during past interglacials. We have conducted this study within that context, and tried to devise a methodology that is general enough to capture the character of different kinds of past interglacial forcing (e.g. warming occurring at different rates, driven more by $CO_2$ or insolation forcing, with or without additional precipitation) so that we can better interpret subglacial observations to say something about the potential ice-sheet geometries they encode. We see this as a very practical endeavor because these samples exist, and are being used to try to understand past ice-sheet contributions to sea level, but they are often not yet precisely dated and the forcing is unknown. Our work addresses the need to use glaciological models with an accounting for some uncertainties in order to make sense of these observations.**

**We hope that the reviewer will find in lines 227–231 and 355–358 changes to the manuscript which engage with their skepticism about the possibility of using schematic, non-specific forcings to say something meaningful about ice-sheet vulnerability.**

Page: 4

l118. "that have been established"

**References have been added here to motivate the ensemble design. The text now reads:**

l119. "represent the major uncertainties"

A bit too confident here. What about the temperature lapse rate, for example? Can you confidently say that you have sufficiently sampled climate uncertainty, when your forcing isn't even specific for a given period?

**The text has been reworded to be less confident. The temperature lapse rate is included in our model (see lines 205–208). We are not sampling all forms of climate uncertainty, but we have designed our ensemble so that we sample major well-described modes of variability that are relevant to the past. See also lines 227–231 in the revised manuscript about the uncertainty in the climate forcing: These timeslices thus represent two end-members: lower $CO_2$/high insolation (HTM) and higher $CO_2$/lower insolation (preindustrial). In this way, they are representative of two known modes of interglacial warmth, and capture both the spatial and seasonal patterns associated with them. However, they do not capture different spatial/seasonal patterns that might be associated with climates warmer than modern, and we assume both spatial and seasonal patterns stay fixed as we conduct the interglacial warming experiments.**

**Page: 6**

l153. "Thickness differences are shown"

Should also show red where there is no ice observed but modelled. Contour the observed (BM) ice sheet extent rather than masking the results with it.

**The figure has been updated following feedback from both reviewers. Thickness differences are shown on the model grid so that where the is modelled ice but no observed ice, this is now visible. The BedMachine mask now appears as a black contour in each panel.**

l156. "Figure 2 shows the method that we apply to calculate sea-level potential"

This is also described in 2.3. why split the description between here and there?

**We have moved this text and combined it with section 2.3 so that the description of our method for calculating sea level potential is in one place. See lines 290-303 in the revised manuscript.**

Page: 7

l178. "most sensitive to HTM climate"

Not sure what that means. Most sensitive compared to what? How can this be used to constrain the ensemble?

**We have adjusted this text to be more clear. The revised text reads, "In our ensemble, the sea-level potential for this site is most sensitive to spatial climatology, because knowing that parameter with certainty would reduce the spread of the ensemble by the greatest amount."**

**This can be used to constrain the ensemble because, for instance, if we knew that the site was ice-free during the Holocene Thermal Maximum (or a time period that was similar to the Holocene Thermal Maximum, e.g. with a stronger insolation forcing and a weaker $CO_2$ forcing) then we would be able to exclude the ensemble members using other spatial climatologies, and ice-free conditions at that site at that time would be associated with a narrower spread of possible ice-sheet geometries and thus sea level contributions.**

Page: 8

l184. Some more modelling information is needed.

How does the model solve for the thermal state? What geothermal heat flux and surface boundary conditions are used. How is isostatic adjustment calculated? Does the model have a name?

Grid resolution could also be mentioned here.

**The requested details have been added to section 2.1 of the manuscript. This section now includes the following text: We used a three-dimensional thermomechanical ice sheet model (Penn State Ice-Sheet Model).... We solve for the thermal state of the ice-sheet by considering vertical diffusion of geothermal heat from below (constant geothermal flux of 42 mW m$^{-2}$) and above (surface climate), heat generated by internal friction and friction at the basal boundary, and advection from new accumulation and ice-flow. Isostatic adjustment is calculated using an elastic lithosphere-relaxing asthenosphere model, where the response time of the bedrock to a changing ice and ocean load is a free parameter that we vary in our ensemble... For our experiments, we calculate basal sliding coefficients through an inversion with modern climate forcing everywhere beneath the ice-sheet; ice-free areas are then assigned a sliding coefficient that reflects one of two end-members based on their elevation, with bedrock below sea-level assigned a weak bed and subaerial bedrock assigned a strong bed... We use a linear temperature lapse-rate correction of 5 $^o$C km$^{-1}$ (Abe-Ouchi et al. 2007) to downscale the 40 km$^2$ climate forcing to the 10 km$^2$ ice-sheet grid, and to dynamically adjust the ice-sheet surface temperature as the ice geometry evolves (e.g. height-mass balance feedback, Weertman 1961). Surface melt is calculated as 5 mm of melt per positive degree day.**

l190. "sliding to reduce the mismatch between the modelled and observed ice-sheet geometry"

Is this done here? How does that work for an initial state at LGM?

**We have revised the text to explain how this works for an initial state at LGM:** For an initial state at LGM, the ice margin advances onto the submarine continental shelf with a relatively weak bed (Fig 1b).

l204. "exist continuously for the last 21 kyr"

Reference needed. At what spatial and temporal resolution? How do these compare to state-of-the-art

regional climate model simulations for the present?

**This text has been changed to be clearer about what kinds of reconstructions are available and their utility for generating SMB fields to force ice-sheet models: Climate reconstructions exist continuously for the last 21 kyr at a range of spatial and temporal resolutions, because during this period ice cores, climate models and modern observational data overlap (Buizert et al. 2018, Badgeley et al. 2020, Osman et al. 2021). These reconstructions can be used to infer past patterns of surface mass balance to force ice-sheet models. Conversely, for most past warm periods before 21 ka there is little known about the precise patterns of SMB.**

**For the reconstruction that we use, the preindustrial climatology is forced to be closed to a reanalysis product that includes information from meteorological stations and ice core records (see lines 282-287 in the revised manuscript for the description we have added in response to this comment). All reconstructions are lower spatial and temporal resolution that state-of-the-art regional climate model simulations for the present, but have the advantage that they have accounted for changes in boundary conditions like insolation which are not relevant to near-term regional climate change but were important for driving past periods of warmth.**

l206. "the precise patterns"

Are you sure you know the precise pattern after 21 ka? Clarify and compare to RCMs at present.

**We have adjusted the text to avoid the implication that we know the precise pattern after 21ka. It now reads (lines 220–221): "Conversely, for most past warm periods before 21 ka, we have less confidence in high-resolution climate reconstructions." Spatiotemporal patterns of climate change during the last deglaciation have been addressed in other work (Buizert et al. 2018, Tabone et al. 2024) and are beyond the scope of the present manuscript, especially given we focus on the results using simulations initialized either with LGM forcing or initialized to match the modern ice-sheet as closely as possible. This section is meant to show that the overall results do not change substantially when replacing the modern spin-up case with an experiment that branches off following the last deglaciation.**

**The reconstruction (averaged over 1955–1995; panel "a" below) differs from the RACMO2.3 Climatology (averaged over 1960–1990; panel "b" below; Noël et al. 2015) by less than 2ºC over the entire ice sheet (see below). The reconstruction is slightly colder around the ice-sheet margins and slightly colder in the interior compared to RACMO2.3. Given that our experiments use warming rates between 1º to 2º C per 1,000 years, and even marginal sites take thousands of years to deglaciate (Figure 2), these differences are minor in the context of our experiments.**

[Figure]

**Comparison of surface temperature in the reconstructed climatology (a) and RACMO2.3 (b) for the period of 1950s–1990s. The difference (reconstruction minus RACMO) is shown in panel c, demonstrating less than 2ºC difference over the area covered by the Greenland ice sheet (black contour is BedMachine ice mask; Morlighem et al. 2017).**

l211. "Both forcings come from a blended model-data reconstruction"

A bit more detail would be good here beyond just adding the reference. What data and models are involved here?

**We have added a more thorough description of the methodology used to produce the spatial climate reconstructions in lines 282-287: The reconstruction combines a reanalysis product that uses meteorological station records and regional climate model output (Box et al. 2009), three ice-core-based temperature reconstructions (Buizert et al. 2014), and a transient simulation of the last deglaciation (Liu et al. 2009) to produce a high-resolution, transient reconstruction of the last 21kyr; a complete description of the methods used can be found in Buizert et al. (2018).**

**We have added text in this section referring the reader to section 2.2.5 for this more detailed description (lines 231–233): Both forcings come from a blended model-data reconstruction that includes seasonally resolved spatial and temporal variability (Buizert et al. 2018) downscaled to 40km resolution (Figure 1, Section 2.2.5).**

Page: 9

l213. "Lapse rate applied to precipitation"

Is there no SMB-height feedback parametersed? This may be an important feedback as well.

**Yes, We use a linear temperature lapse-rate correction of 5 ºC km⁻¹ (Abe-Ouchi et al. 2007) to downscale the 40 km² climate forcing to the 10 km² ice-sheet grid, and to dynamically adjust the ice-sheet surface temperature as the ice geometry evolves (e.g. height-mass balance feedback, Weertman 1961). This text has been added to the revised manuscript (Lines 205-208).**

l214. "periods can be volumetrically by increased"

Word missing?

**Thank you for pointing this out – the word "offset" was missing. This sentence now reads, "Ice-sheet margin retreat inland of its present-day position during past warm periods can be volumetrically offset by increased accumulation in precipitation-limited areas whose temperature remains far below the melting point." (Lines 235–236).**

l222. "not applying precipitation lapse-rate correction, because there is no compensation for increasing melt."

Not clear what is meant here.

**We have clarified this sentence in the revised manuscript (Lines 242-244): Simulations where the precipitation-lapse rate correction is ignored do not have a mechanism for increased precipitation to offset ice loss as temperature rises and melting increases, so these ensemble members are expected to predict greater total ice-sheet mass loss prior to deglaciation.**

l233. "an interglacial warming ramp"

So the forcing is schematic? Maybe better not to call it interglacial?

Also, ice sheet shrinking during the LIG and MIS11 was substantial during peak interglacial conditions, where the forcing is stabilised or even already decreasing. See also main comment on realism of ramping up forcing and possible implications.

**We have added "idealized" to emphasize that the interglacial forcing is schematic (line 255). By using a range of interglacial warming rates, we have captured how GIS change depends on the rate of interglacial warming. In this work, we are not looking at steady-state (stable forcing) or glacial advance (decreasing forcing) but we appreciate this comment from the reviewer and believe that this should be a focus of future work (see revised manuscript lines 464-467 and 474–481, which reflect this). See also our response to main comment on realism of the applied forcing, paraphrased here: The advantage of our approach is that we are interested not in a specific time but in patterns that can reveal that shared vulnerabilities across different realizations of a deglaciation. This is useful for the present context, because this work is motivated by the increasing prevalence of studies where subglacial material is being leveraged to learn about past ice-free conditions without knowing the precise timing of when those conditions occurred. We propose this framework as a different and complementary way to think about model-data integration.**

l239. "use two mantle relaxation times"

What kind of bedrock model is used here? More detail needed.

**More information about the bedrock model has been added to revised manuscript section 2.1:**
Isostatic adjustment is calculated using an elastic lithosphere-relaxing asthenosphere model, where the response time of the bedrock to a changing ice and ocean load is a free parameter that we vary in our ensemble.

Page: 10

l257. "an evolving climatology for 21kyr"

This needs more details. What models and data are used? This is described in Buizert et al. but the most important basics should become clear without visiting the reference.

**We have added the following details on the reconstruction in lines 282-287: The reconstruction combines a reanalysis product that uses meteorological station records and regional climate model output (Box et al. 2009), three ice-core-based temperature reconstructions (Buizert et al. 2014), and a transient simulation of the last deglaciation (Liu et al. 2009) to produce a high-resolution, transient reconstruction of the last 21kyr; a complete description of the methods used can be found in Buizert et al. (2018).**

l258. "ice sheets differ in their geometry"

Add reference to 1b?

**Added.**

l261. "sea-level potential, defined"

The first definition would be correct for one specific ice sheet trajectory. Say so!

The second definition holds for an ensemble of runs. Say so!

Would it be possible to have just one definition instead?

A bit awkward to evoke the width of the histogram for this definition. Isn't it just the ensemble median value?

**We thank the reviewer for these clarifying insights. We have revised this section to be more explicit about our definition for sea-level potential, as well as how it applies to a specific ice-sheet trajectory versus our ensemble. The revised manuscript now reads (lines 291-295): "For each 10 km model grid cell, we analysed the ensemble to find the first timestep the site becomes ice-free in each simulation (Figure 2a). For the first ice-free timestep in each simulation, we save the ice-sheet volume and extent, and collate these values across the whole ensemble (Figure 2b,c). We define the median of the ice-sheet volume histogram for each location as the sea-level potential for that location. For each site, we consider the uncertainty to be the ensemble spread of sea-level potential. In our method, the sea-level potential is specific to each site and to our ensemble."**

l265. "or the x-axis of Figure 2b"

What does the 'or' link to here? The parameter, the Greenland contribution? In either case, the x-axis is not the same as those items. Reformulate!

**This text has been reformulated to be more clear. The text now reads (lines 298–303): We define the sensitivity to each parameter as the subensemble spread for each parameter separately divided by the full ensemble spread. When these spreads are equal, the sensitivity score is one and the ensemble spread does not change if we consider only a particular value of that parameter. When a parameter spans a smaller range than the full ensemble does, this number is less than one, indicating that knowledge of this parameter reduces the spread of sea level potential. The parameter that each location is most sensitive to is determined by identifying the smallest sensitivity**

**score, and the relative importance of each parameter is determined by the ranking the scores from lowest to highest (Figure 3).**

l265. "the parameter that the GIS responds most sensitively to"

This obviously depends on the range the parameter is sampled at, which makes this whole concept problematic. Even more so for parameters that are categorical. The parameters are not sampled in a statistically meaningful way, which makes these inferences difficult.

**This text has been removed in the revised manuscript, and reworded to be clear that the sensitivity we are referencing is specific to our ensemble (lines 296–298): Repeating this process for each grid cell, we also look at our results along every dimension of our ensemble, enabling us to calculate the importance of each parameter for each site and rank which of the considered parameters dominate the ensemble spread at each location.**

**The reviewer makes a fair point, and goes to their main comment that we have not fully sampled the full uncertainty in past climate forcing. However, we reiterate from our response above that we have made specific assumptions about the model set-up and ensemble design to look at shared characteristics of deglaciation scenarios driven by forcings that represent Pleistocene interglacials, but are not particular to one time period. Our approach is certainly not perfect, but we disagree that it "makes the whole concept problematic." We think that the strength is the concept is that, when the results are interpreted in the context of the assumptions and choices that are made in setting up the ensemble, it can help us understand which kinds of uncertainty dominate the ensemble spread of sea level potential in different regions, which we think will be of interest to different communities. As mentioned previously, we have revised the manuscript so that these assumptions are made more clear (see for example, lines 229–231, 364–365, and 431–435, and 477–479).**

l266. "dividing the width of the histogram"

Can this be formulated in statistical terms, rather than using the image of the histogram? Isn't this just the (sub-)ensemble spread of the potential?

**Yes, this is the subensemble spread – we have reworded this (line 298) to use the reviewer's suggestion.**

Page: 11

l278. "the distribution of ice volume estimates"

The term doesn't speak to me (see elsewhere), but for consistency: refer to the potential here?

**The Figure caption has been revised as follows: "A) Shows which ensemble parameter exerts the strongest control on the ensemble spread of sea-level potential."**

l283. "the first regions to deglaciate in our ensemble"

Difficult to imagine that wouldn't be the case in any other ensemble. Did an ice sheet ever deglaciate starting from the centre?

**We understand that this may be obvious but we have chosen to keep this text because we believe the reiteration of what sea-level potential means by applying it specifically to our ensemble helps reinforce our definition and orient the reader to our results.**

l284. "the last regions to deglaciate in our ensemble"

Can the behaviour of the ensemble be illustrated in some form? Maybe a figure like 2a for the entire ice sheet volume and 2d maps for a high, medium and a low sensitivity member of a few time slices.

**We appreciate this suggestion from the reviewer. We have created another figure to illustrate the behavior of the ensemble. This is included in the new manuscript as Figure 5.**

Page: 12

l296. "the mean amount"

I think in 2.3 this was defined as the median?

**The caption has been corrected to reflect that it is indeed the median.**

l297. "width of the full histogram"

= ensemble spread?

**Changed "width of the full histogram" to "ensemble spread" in Figure 4 caption.**

l312. "(Figure 4a)"

A bit difficult to understand. Maybe best placed right, as it is a combination of b) and c). So we can try to understand those first as we visit the figure.

**This figure has been updated following comments from both reviewers. The simplified version omits the panel "a" in favor of just showing the old panels "b" and "c," which we agree is more clear.**

l312. "This map reveals"

Can you explain why this is an interesting target to look at? I understand looking for regions with low error, but why is it interesting to look at low potential? I would imagine high potential and low error to be an interesting target for constraining deglaciations in the past.

**Low-potential areas represent the places we expect the ice-sheet to deglaciate first, which is a key target for subglacial drilling campaigns seeking to constrain the geometry of the ice sheet at time periods in the past when the ice sheet was smaller than today but not completely deglaciated. High potential and low error indeed represent complete deglaciation events, which are also interesting. We think this is a compelling aspect of the sea-level potential framework, because it represents a way to compare a wide range of data-based observations with an ensemble of ice-sheet model outputs to address separate but complementary hypotheses about past ice-sheet change.**

Page: 13

l319. "(Figure 3a)"

My understanding is that the result of this analysis is highly model dependent. Already the choice of parameters is subjective and may not be the same for another model. This should come out clearly in the discussion, but better be mentioned already here.

**We added the phrase "in our ensemble" in the first sentence of this paragraph to be more clear that this is a result of the present study (line 338–339). We have added a further discussion of the model and ensemble dependence of our results in lines 355-358.**

l335. "where ice-cover at that site is associated with a wide range of potential ice-sheet geometries"

I am not convinced about this point. What you are showing is the opposite: ice-free conditions associated with a wide range of potential. What makes you think the opposite is also true?

**We have adjusted the language to reflect that we are indeed analyzing the range of geometries consistent with ice-free conditions, as the reviewer points out. The text now reads (lines 358–361): Nevertheless, in this work, we find that areas with a wide spread of sea level potential (e.g. Camp Century; Christ et al. 2023) can be thought of as places where ice-free conditions at that site is associated with a wide range of potential ice-sheet geometries; the ice sheet can grow and shrink in different regions while ice remains at that site.**

**The reviewer makes a good point that we do not address in this manuscript – the way that the ice sheet grows from an deglaciated state to a (re)glaciated state is not well known. We suggest that this should be a target of future work, and have referenced this in the discussion (lines 472–477): "Due to hysteresis effects…"**

l337. "require a contribution of +1.4 m SLE"

Why not formulate this as potential of x and spread of y?

**We have reformulated this sentence as suggested by the reviewer (lines 361–364): For example, we find sea level potential for the Camp Century site is 3.2m with a spread of 1.4 – 5.6 m SLE. An analysis of subglacial material from Camp Century found that site was ice-free during MIS 11; our method thus provides a lower bound on the GIS contribution to global sea level at that time (Christ et al. 2023).**

l338. "has contributed +5.6 m SLE (Christ et al. 2023)"

So this is not from your results? A bit confusing. Clarify.

**We have rewritten this sentence so that it is clearer that these are our results (see the previous response).**

l339. "by adding constraints on simultaneously ice-free conditions at more than one location"
Not sure I understand how that would work. Could you explain?

**Consider Figure 2c, where the green dot is a location where we are interested in sea-level potential. The colored contours represent locations that are ice-free at the same time as the location of interest for part of our ensemble, but remain ice-covered when our location of interest is ice-free in other parts of the ensemble. If two subglacial samples from two areas were identified as ice-free at the same time, this would reduce the ensemble spread relative to considering ice-free conditions at only one site (as we have shown in Figure 2).**

l339. "by considering a subset of our parameter space."

You have to distinguish between the actual uncertainty and the uncertainty produced by your ensemble.

Do you want to reduce uncertainty in your ensemble by removing ensemble members that disagree? Sure, you could remove a lot of uncertainty by just running one single representation. Then uncertainty is zero. But that doesn't change anything about the actual uncertainty.

**The uncertainty produced by the ensemble is a representation of the actual uncertainty, because we have designed the ensemble to capture major modes of interglacial climate variability. It is not a perfect representation of the actual uncertainty, because there is still room for improvement. We have updated the revised manuscript to be more explicit that the sea level potential and its associated uncertainty may be model and ensemble dependent (lines 355–358): "We note that sea-level potential, and its associated uncertainty…"**

**We do not want to "reduce uncertainty in our ensemble by removing ensemble members that disagree," we want to use our ensemble approach to identify which parameters result in the most divergent deglaciation pathways at every location, so that we can understand which parameters are most important to constrain when interpreting evidence for ice-free conditions.**

Page: 14

l366. "the elevation-surface mass balance feedback"

Not clear how that is included in the current modelling.

**Section 2.1 has been revised to address this. Lines 205-208 now read: We use a linear temperature lapse-rate correction of 5 ℃ km$^{-1}$ (Abe-Ouchi et al. 2007) to downscale the 40 km$^2$ climate forcing to the 10 km$^2$ ice-sheet grid, and to dynamically adjust the ice-sheet surface temperature as the ice geometry evolves (e.g. height-mass balance feedback, Weertman 1961).**

Page: 15

l390. "Our approach circumvents the need for direct reconstruction"

I don't believe that you have circumvented the problem. A different climate forcing will lead to a different deglaciation and therefore a different distribution of potential than what you are finding. You seem to assume that the two climatologies in combination with the Buizert forcing for the last deglaciation is sufficient to generate climate forcing that could represent any deglaciation? I don't think that is true. See also general comment.

**We agree, circumvent is too strong a word and not the meaning that we were intending to get across. We have revised the text so that it now reads: "Our approach avoids direct reconstruction of SMB for a particular interglacial by considering a range of plausible forcings"**

**We disagree with the reviewer on some aspects of this general comment, as outlined earlier in the response. Paraphrasing that earlier response here, our approach focuses on identifying patterns of shared vulnerabilities across deglaciation scenarios rather than specific time periods. This is crucial for interpreting subglacial materials, which often lack precise timing, to infer past ice-free conditions. By addressing uncertainties in past climate forcing, we developed a general methodology to interpret various interglacial forcings (e.g., $CO_2$, insolation, precipitation) and their impact on ice-sheet geometries. This practical framework helps make sense of subglacial samples and their implications for past ice-sheet contributions to sea level, despite dating and forcing uncertainties. We share the reviewer's opinion that our forcings are imperfect and we hope that future work can improve on these important constraints.**

l394. "may become dominant in the future as boundary conditions and forcings evolve"

More relevant for this study and its potential application (identifying drill sites) is whatever happened in the past during LIG and MIS11. I don't understand the focus on the future here. See also general comment.

**We have chosen to keep this reference to the future in the revised manuscript. This sentence is a reflection on how the approach could be improved in order to capture kinds of climate variability that are currently not captured, which we think is in the spirit of many of the reviewer's comments. This comment and the next two are all hinge on the general idea that the past tells us something about the future, which we have addressed in the general comment section, and paraphrase again below.**

l397. "correctly predicting the spatial patterns of climate over Greenland"

Again, not clear why this discussion point is about the future. We have methods to project future sea-level contributions that are far more advanced than this modelling approach. The need of subglacial observations is to constrain the past, not the future, I suppose.

**The reference to the future has been removed, so that this sentence now reads: Nevertheless, our results confirm the primacy of correctly predicting the spatial patterns of climate over Greenland (Edwards et al. 2014) for inferring past sea-level change, and suggest that selecting sites that have lower uncertainty in their sea-level potential will increase the impact of subglacial observations.**

l398. "first meter of future sea-level change"

This is the wrong question. See general comment.

**This sentence has been revised; see previous response. As a general response to the past three comments, we reiterate that whether the past is representative of the future is the subject of much debate, but efforts to better understand how the ice sheet responded to climate change in the past to provide context for ongoing changes and potentially inform projections for the future are useful. Throughout the text, we have reduced emphasis on the future in response to the reviewer's comments, so that it focuses more closely on the past.**

Page: 16

l411. "the patterns that hold true regardless of the style of deglaciation"

I disagree with this statement. Ignoring the specific forcing does not render the results more general and true. The presented results are just the specific response to another specific forcing.

**We have revised this sentence so that it now reads: This allows us to overcome the challenges associated with perfectly simulating a particular time period in favor of identifying the patterns that are common among different styles of deglaciation.**

**We have addressed the comment about the results being general in other parts of the response.**

l412. "useful insight into the uncertain future of the GIS"

**We have chosen to keep this reference to the future because it is generic and speaks to the overarching idea that an improved understanding of the past can inform our projections for the future.**

l414. "the first few meters of sea-level rise from Greenland are likely to originate"

Not clear to me how your method is providing that information. There is mass loss happening in places that deglaciate late or never in your simulations.

**We appreciate this feedback. Indeed our simulations account for mass lost in places that have not yet deglaciated, which is one of the reasons that the sea level potential has a spread associated with it, in addition to changes in the margin position. We have revised the text in response to this feedback so that it now reads (lines 459-461): Our approach complements far-field sea-level records by providing a method to quantify a minimum sea-level contribution for the GIS during periods when directly dated subglacial samples are available, regardless of the final geometry of the ice sheet at the time of maximum retreat during an interglacial (e.g. Dyer et al. 2021, Barnett et al. 2023).**

l425. "can provide robust estimates"

Robust against what? Are you accounting for all of the uncertainties in your modelling?

**We have removed the word robust. The sentence now reads (lines 470–472): At present, the ensemble has been designed specifically to demonstrate how subglacial material documenting past ice-free conditions, in combination with numerical ice-sheet modelling, can provide estimates of continent-wide sea level contribution that account for dominant sources of uncertainty.**

**We are not seeking to account for all of the uncertainties, but rather to demonstrate a way that data–model integration might happen in a mutually beneficial way. We think the sentence now better reflects the uncertainties that we are trying to capture.**

Page: 17

l442. "into information that can inform adaptation efforts"

Adaptation to sea-level rise? Not clear to me what the proposed translation does.

**We have clarified this sentence to reflect that we are talking about efforts to adapt to sea-level change.**

l443. "we expect that improved knowledge of the past spatial mass balance patterns and relationships between temperature and precipitation change will have the greatest impact"

If that is true, which I believe, your method is not general at all.

**I think we disagree about a method being general and results being general. Our results are specific to a particular model set-up and ensemble design, which we have repeated here (lines 481–482). We have endeavored to make that ensemble design general insofar as it is not tied to a particular interglacial and can thus be used to identify behaviors that are common among a range of deglaciation events. I agree that our results are only "general" insofar as they are valid as long as the assumptions that we have made hold true. We have clarified and further articulated the question of model and ensemble dependence in the revised manuscript (see lines 355-358). What we are communicating in this concluding section is that the method itself can be used in the future as our knowledge of past boundary conditions improves, and that process of refinement can be used to make better assumptions and accordingly have greater confidence in the general applicability of the results. I think we are in agreement that the sentence is true, and we have tried in lines 355-358 to better bridge the gap between the method being general (and flexible to future improvements) and the results being general.**

l453. "6 Code Availability"

If you want the described framework to be used by anyone else, this is the place to share the code. Making work reproducible by others is becoming a standard throughout the community. I strongly suggest to think about how to make your work accessible and reproducible.

**The Code availability section has been revised to read: The scripts needed to reproduce the results of this manuscript are available at https://github.com/bkeisling/sea-level-potential. This work employs the ice sheet model code described in DeConto et al. (2021).**

l456. "7 Data Availability"

The same as for code availability above. What data is needed to reproduce these results of this paper?

**The data availability section has been revised to read: Ice-sheet model output in the form of NetCDF files is available from the Arctic Data Center under the title "Idealized Greenland Ice Sheet Deglaciation Experiments." A doi for these data will be provided upon publication.**

References

Abe-Ouchi, A., Segawa, T., and Saito, F.: Climatic conditions for modelling the Northern Hemisphere ice sheets throughout the ice age cycle, Climate of the Past, 3, 423–438, 2007.

Aschwanden, A., Fahnestock, M. A., Truffer, M., Brinkerhoff, D. J., Hock, R., Khroulev, C., Mottram, R., and Khan, S. A.: Contribution of the Greenland Ice Sheet to sea level over the next millennium, Science Advances, 5, eaav9396, https://doi.org/10.1126/sciadv.aav9396, 2019.

Badgeley, J. A., Steig, E. J., Hakim, G. J., and Fudge, T. J.: Greenland temperature and precipitation over the last 20,000 years using data assimilation, https://doi.org/10.5194/cp-2019-164, 2020.

Barnett, R. L., Austermann, J., Dyer, B., Telfer, M. W., Barlow, N. L. M., Boulton, S. J., Carr, A. S., and Creel, R. C.: Constraining the contribution of the Antarctic Ice Sheet to Last Interglacial sea level, Sci. Adv., 9, eadf0198, https://doi.org/10.1126/sciadv.adf0198, 2023.

Box, J. E., Yang, L., Bromwich, D. H., and Bai, L.-S.: Greenland Ice Sheet Surface Air Temperature Variability: 1840–2007*, Journal of Climate, 22, 4029–4049, https://doi.org/10.1175/2009JCLI2816.1, 2009.

Briner, J. P., Cuzzone, J. K., Badgeley, J. A., Young, N. E., Steig, E. J., Morlighem, M., Schlegel, N.-J., Hakim, G. J., Schaefer, J. M., Johnson, J. V., Lesnek, A. J., Thomas, E. K., Allan, E., Bennike, O., Cluett, A. A., Csatho, B., de Vernal, A., Downs, J., Larour, E., and Nowicki, S.: Rate of mass loss from the Greenland Ice Sheet will exceed Holocene values this century, Nature, 586, 70–74, https://doi.org/10.1038/s41586-020-2742-6, 2020.

Buizert, C., Gkinis, V., Severinghaus, J. P., He, F., Lecavalier, B. S., Kindler, P., Leuenberger, M., Carlson, A. E., Vinther, B., Masson-Delmotte, V., White, J. W. C., Liu, Z., Otto-Bliesner, B., and Brook, E. J.: Greenland temperature response to climate forcing during the last deglaciation, Science, 345, 1177, https://doi.org/10.1126/science.1254961, 2014.

Buizert, C., Keisling, B. A., Box, J. E., He, F., Carlson, A. E., Sinclair, G., and DeConto, R. M.: Greenland-Wide Seasonal Temperatures During the Last Deglaciation, Geophysical Research Letters, 45, 1905–1914, https://doi.org/10.1002/2017GL075601, 2018.

Christ, A. J., Rittenour, T. M., Bierman, P. R., Keisling, B. A., Knutz, P. C., Thomsen, T. B., Keulen, N., Fosdick, J. C., Hemming, S. R., Tison, J.-L., Blard, P.-H., Steffensen, J. P., Caffee, M. W., Corbett, L. B., Dahl-Jensen, D., Dethier, D. P., Hidy, A. J., Perdrial, N., Peteet, D. M., Steig, E. J., and Thomas, E. K.: Deglaciation of northwestern Greenland during Marine Isotope Stage 11, Science, 381, 330–335, https://doi.org/10.1126/science.ade4248, 2023.

Cluett, A. A. and Thomas, E. K.: Summer warmth of the past six interglacials on Greenland, Proc. Natl. Acad. Sci. U.S.A., 118, e2022916118, https://doi.org/10.1073/pnas.2022916118, 2021.

De Vernal, A. and Hillaire-Marcel, C.: Natural Variability of Greenland Climate, Vegetation, and Ice Volume During the Past Million Years, Science, 320, 1622–1625, https://doi.org/10.1126/science.1153929, 2008.

DeConto, R. M., Pollard, D., Alley, R. B., Velicogna, I., Gasson, E., Gomez, N., Sadai, S., Condron, A., Gilford, D. M., Ashe, E. L., Kopp, R. E., Li, D., and Dutton, A.: The Paris Climate Agreement and future sea-level rise from Antarctica, Nature, 593, 83–89, https://doi.org/10.1038/s41586-021-03427-0, 2021.

Dutton, A., Carlson, A. E., Long, A. J., Milne, G. A., Clark, P. U., DeConto, R., Horton, B. P., Rahmstorf, S., and Raymo, M. E.: Sea-level rise due to polar ice-sheet mass loss during past warm periods, Science, 349, aaa4019–aaa4019, https://doi.org/10.1126/science.aaa4019, 2015.

Dyer, B., Austermann, J., D'Andrea, W. J., Creel, R. C., Sandstrom, M. R., Cashman, M., Rovere, A., and Raymo, M. E.: Sea-level trends across The Bahamas constrain peak last interglacial ice melt, Proc Natl Acad Sci USA, 118, e2026839118, https://doi.org/10.1073/pnas.2026839118, 2021.

Edwards, T. L., Fettweis, X., Gagliardini, O., Gillet-Chaulet, F., Goelzer, H., Gregory, J. M., Hoffman, M., Huybrechts, P., Payne, A. J., Perego, M., Price, S., Quiquet, A., and Ritz, C.: Effect of uncertainty in surface mass balance–elevation feedback on projections of the future sea level contribution of the Greenland ice sheet, The Cryosphere, 8, 195–208, https://doi.org/10.5194/tc-8-195-2014, 2014.

Greenbaum, J. S., Blankenship, D. D., Young, D. A., Richter, T. G., Roberts, J. L., Aitken, A. R. A., Legresy, B., Schroeder, D. M., Warner, R. C., van Ommen, T. D., and Siegert, M. J.: Ocean access to a cavity beneath Totten Glacier in East Antarctica, Nature Geoscience, 8, 294–298, https://doi.org/10.1038/ngeo2388, 2015.

Helsen, M. M., Van De Berg, W. J., Van De Wal, R. S. W., Van Den Broeke, M. R., and Oerlemans, J.: Coupled regional climate–ice-sheet simulation shows limited Greenland ice loss during the Eemian, Clim. Past, 9, 1773–1788, https://doi.org/10.5194/cp-9-1773-2013, 2013.

Liu, Z., Otto-Bliesner, B. L., He, F., Brady, E. C., Tomas, R., Clark, P. U., Carlson, A. E., Lynch-Stieglitz, J., Curry, W., Brook, E., Erickson, D., Jacob, R., Kutzbach, J., and Cheng, J.: Transient Simulation of Last Deglaciation with a New Mechanism for Bolling-Allerod Warming, Science, 325, 310–314, https://doi.org/10.1126/science.1171041, 2009.

Morlighem, M., Williams, C. N., Rignot, E., An, L., Arndt, J. E., Bamber, J. L., Catania, G., Chauché, N., Dowdeswell, J. A., Dorschel, B., Fenty, I., Hogan, K., Howat, I., Hubbard, A., Jakobsson, M., Jordan, T. M., Kjeldsen, K. K., Millan, R., Mayer, L., Mouginot, J., Noël, B. P. Y., O'Cofaigh, C., Palmer, S., Rysgaard, S., Seroussi, H., Siegert, M. J., Slabon, P., Straneo, F., van den Broeke, M. R., Weinrebe, W., Wood, M., and Zinglersen, K. B.: BedMachine v3: Complete Bed Topography and Ocean Bathymetry Mapping of Greenland From Multibeam Echo Sounding Combined With Mass Conservation: BEDMACHINE GREENLAND V3, Geophysical Research Letters, https://doi.org/10.1002/2017GL074954, 2017.

NEEM community members: Eemian interglacial reconstructed from a Greenland folded ice core, Nature, 493, 489–494, https://doi.org/10.1038/nature11789, 2013.

Osman, M. B., Tierney, J. E., Zhu, J., Tardif, R., Hakim, G. J., King, J., and Poulsen, C. J.: Globally resolved surface temperatures since the Last Glacial Maximum, Nature, 599, 239–244, https://doi.org/10.1038/s41586-021-03984-4, 2021.

Past Interglacials Working Group of PAGES: Interglacials of the last 800,000 years, Rev. Geophys., 54, 162–219, https://doi.org/10.1002/2015RG000482, 2016.

Pelle, T., Greenbaum, J. S., Ehrenfeucht, S., Dow, C. F., and McCormack, F. S.: Subglacial Discharge Accelerates Dynamic Retreat of Aurora Subglacial Basin Outlet Glaciers, East Antarctica, Over the 21st Century, JGR Earth Surface, 129, e2023JF007513, https://doi.org/10.1029/2023JF007513, 2024.

Plach, A., Nisancioglu, K. H., Le clec'h, S., Born, A., Langebroek, P. M., Guo, C., Imhof, M., and Stocker, T. F.: Eemian Greenland SMB strongly sensitive to model choice, Climate of the Past, 14, 1463–1485, https://doi.org/10.5194/cp-14-1463-2018, 2018.

Robinson, A., Alvarez-Solas, J., Calov, R., Ganopolski, A., and Montoya, M.: MIS-11 duration key to disappearance of the Greenland ice sheet, Nat Commun, 8, 16008, https://doi.org/10.1038/ncomms16008, 2017.

Tabone, I., Robinson, A., Montoya, M., and Alvarez-Solas, J.: Holocene thinning in central Greenland controlled by the Northeast Greenland Ice Stream, Nat Commun, 15, 6434, https://doi.org/10.1038/s41467-024-50772-5, 2024.

Weertman, J.: Stability of ice-age ice sheets, J. Geophys. Res., 66, 3783–3792, https://doi.org/10.1029/JZ066i011p03783, 1961.